# Orthogonal Survival Learners for Estimating Heterogeneous Treatment Effects from Time-to-Event Data

**Dennis Frauen,**\* **Maresa Schröder,**\* **Konstantin Hess,** **Stefan Feuerriegel**
LMU Munich
Munich Center for Machine Learning
{frauen, maresa.schroeder, k.hess, feuerriegel}@lmu.de

## Abstract

Estimating heterogeneous treatment effects (HTEs) is crucial for personalized decision-making. However, this task is challenging in *survival analysis*, which includes time-to-event data with censored outcomes (e.g., due to study dropout). In this paper, we propose a toolbox of *orthogonal* survival learners to estimate HTEs from time-to-event data under censoring. Our learners have three main advantages: (i) we show that learners from our toolbox are guaranteed to be orthogonal and thus robust with respect to nuisance estimation errors; (ii) our toolbox allows for incorporating a custom weighting function, which can lead to robustness against different types of low overlap, and (iii) our learners are *model-agnostic* (i.e., they can be combined with arbitrary machine learning models). We instantiate the learners from our toolbox using several weighting functions and, as a result, propose various neural orthogonal survival learners. Some of these coincide with existing survival learners (including survival versions of the DR- and R-learner), while others are novel and further robust w.r.t. low overlap regimes specific to the survival setting (i.e., survival overlap and censoring overlap). We then empirically verify the effectiveness of our learners for HTE estimation in different low-overlap regimes through numerical experiments. In sum, we provide practitioners with a large toolbox of learners that can be used for randomized and observational studies with censored time-to-event data.

## 1  Introduction

Estimating heterogeneous treatment effects (HTEs) is crucial in personalized medicine [10, 15]. HTEs quantify the causal effect of a treatment on an outcome (e.g., survival), conditional on individual patient characteristics (e.g., age, gender, prior diseases). This enables clinicians to make individualized treatment decisions aimed at improving patient outcomes. For example, knowing the HTE of an anticancer drug on patient survival can inform treatment decisions that are tailored to a patient's unique medical history, thereby maximizing the probability of survival.

A common setting in medicine is *survival analysis* [26, 51]. Survival analysis is aimed at modeling medical outcomes, particularly in cancer care, where the dataset involves *time-to-event data* [54]. That is, the outcome of interest is a time to an event that we are interested in maximizing. For example, in cancer care, a treatment should maximize the time until death or tumor progression [24, 43]. Hence, this requires tailored methods for HTE estimation from time-to-event data.

The fundamental problem that distinguishes survival analysis from standard causal inference is *censoring* [47]. Censoring refers to the phenomenon that some individuals may not have experienced

---

\*Equal contribution.

39th Conference on Neural Information Processing Systems (NeurIPS 2025).

the event (e.g., death, recovery) by the end of the study period (also called right-censoring). For example, older patients may be more likely to drop out of medical studies [35]. Censoring thus requires custom methods to allow for unbiased causal inference. For example, if we simply remove censored individuals from the data, the remaining population may be younger on average, which may lead to biased treatment effect estimates (if the treatment effect is, e.g., different for older patients). As a result, standard methods for HTE estimation (e.g., [23, 33, 46]) are *biased* when used for censored time-to-event data.

In comparison to uncensored data, estimating HTEs from censored time-to-event data is thus subject to three additional *main challenges*: ① **Complex confounding**: Confounders might not only affect treatment and outcome, but also the event and censoring times. Thus, properly adjusting for these confounders is necessary to obtain valid treatment effect estimates. ② **Estimation complexity**: Methods that adjust for confounding under censoring require estimating additional *nuisance functions* over multiple time steps, such as hazard functions. ③ **Different types of overlap**: A necessary requirement for estimating HTEs is sufficient *treatment overlap*, i.e., that every individual has a positive probability of receiving or not receiving the treatment [e.g., 8, 32]. However, under censoring, additional overlap conditions are required, which we refer to as *censoring overlap* and *survival overlap*. For example, every individual must have a nonzero probability of being uncensored (i.e., experiencing the event).

Existing methods have limitations because of which they cannot deal with all the above challenges. For example, for uncensored data, state-of-the-art methods for estimating HTEs are *Neyman-orthogonal meta-learners* (such as the DR- and the R-learner) [e.g., 23, 33, 46]. These have been extended to the survival setting to address challenges ① and ② from above, particularly via survival versions of the DR- and R-learners [7, 52]. However, these learners lack the ability to address challenge ③ of different types of overlap and, as a result, exhibit a large variance under low censoring or survival overlap (e.g., if certain individuals almost never experience the event).

In this paper, we address the above limitations by proposing a novel, general toolbox with Neyman-orthogonal meta-learners for estimating HTEs from censored time-to-event data. Our proposed meta-learners address the challenges from above as follows: ① they use orthogonal censoring adjustments, which enable unbiased and robust estimation under both confounding and censoring; ② they are model-agnostic in the sense that they can be instantiated with any machine learning method (e.g., neural networks) to effectively learn nuisance functions; ③ In contrast to existing survival learners, they effectively overcome the difficulty of treatment effect estimation in the presence of lack of any of the three overlap types through targeted re-weighting of the orthogonal losses.

Our **contributions**[2] are: (1) We propose a novel toolbox with orthogonal learners for estimating HTEs from time-to-event data. Our toolbox allows the specification of a custom weighting function for robust estimation under low treatment-, survival-, and/or censoring overlap. We also provide several extensions of our toolbox to different settings (continuous time, marginalized effects, different causal estimands, and unobserved ties) in our Appendix. (2) We provide theoretical guarantees that learners constructed within our framework are orthogonal and that our learners provide meaningful HTE estimates, regardless of the chosen weighting function. (3) We instantiate our toolbox for several weighting functions and obtain various novel *orthogonal survival learners*. These learners are model-agnostic and can be used in combination with arbitrary machine learning models (e.g., neural networks).

## 2   Related work

Below, we review key works aimed at orthogonal learning for HTEs, especially for censored data. We provide an extended literature review in Appendix A.

**Orthogonal learning of HTEs from uncensored data.** Several meta-learners for HTE estimation have been introduced in the literature, particularly for conditional

Table 1: **Overview of key orthogonal learners** whether they can adjust for censoring / different types of overlap.

| Learner | Censoring | Treat. overlap | Cens. overlap | Surv. overlap |
|---|---|---|---|---|
| DR-learner [46, 23] | ✗ | ✗ | ✗ | ✗ |
| R-learner [33] | ✗ | ✓ | ✗ | ✗ |
| Survival-DR-Learner [32, 52] | ✓ | ✗ | ✗ | ✗ |
| Survival-R-Learner [52] | ✓ | ✓ | ✗ | ✗ |
| **Ours** | ✓ | ✓ | ✓ | ✓ |

[2]Code available at `https://github.com/m-schroder/OrthoSurvLearners`.

average treatment effects [e.g., 9, 28]. Among them, the DR-learner [23, 46] and the R-learner [33] are often regarded as state-of-the-art because these are *orthogonal*, meaning they are based upon semiparametric efficiency theory [3, 48] and robust with respect to nuisance estimation errors [5]. Furthermore, orthogonality typically implies other favorable theoretical properties, such as quasi-oracle efficiency and doubly robust convergence rates [11]. Recently, [32] showed that the R-learner can be interpreted as an overlap-weighted version of the DR-learner, thus addressing instabilities and high variance in low treatment-overlap scenarios.

Orthogonal learners have also been proposed for other causal quantities, such as the conditional average treatment effect on the treated [29], instrumental variable settings [12, 44], HTEs over time [13], partial identification bounds [34, 41], or uncertainty quantification of treatment effect estimates [1]. However, *none* of these learners are tailored to time-to-event data, and, hence, they are *biased* under censoring.

**Model-based learning of HTEs from censored data.** There is some literature for estimating HTEs from time-to-event data that has focused on *model-based learners*, i.e., learners based on specific machine learning models [18]. Examples include tree-based learners [e.g., 7, 16, 45, 54] or neural-network-based learners [8, 21, 40]. Note that these learners are neither model-agnostic (i.e., cannot be used with arbitrary machine learning models) nor orthogonal. Furthermore, model-based learners typically estimate the target HTE via a plug-in fashion and thus suffer from so-called *plug-in bias* [22]. In contrast, (model-agnostic) orthogonal learners remove plug-in bias by fitting a second model based on a Neyman-orthogonal second-stage loss. Nevertheless, model-based learners can be combined with model-agnostic orthogonal learners for the first stage (nuisance estimation).

**Orthogonal learning of HTEs from censored data.** Few works have proposed (orthogonal) meta-learners tailored to censored time-to-event data. Xu et al. [53] introduce an adaptation of existing learners to time-to-event data based on inverse probability of censoring weighting [27, 47]. However, the proposed learner is *only* applicable to experimental data from randomized controlled trials and is sensitive to overlap violations. Gao et al. [14] propose orthogonal learners based on exponential family and Cox models, but *not* neural networks. Xu et al. [52] develop censoring unbiased transformations for survival outcomes; i.e., to convert time-to-event outcomes to standard continuous outcomes, which can then be combined with existing orthogonal learners for estimating HTEs in the standard setting. However, the corresponding survival versions of the DR-learner and R-learner are not robust against survival or censoring overlap violations.

**Research gap:** So far, existing orthogonal survival learners fail to account for different types of overlap violations (see Table 1). To the best of our knowledge, we are thus the first to provide a general toolbox that includes custom weighting functions to ensure robustness against different types of overlap violations (such as survival or censoring overlap violations).

# 3 Problem setup

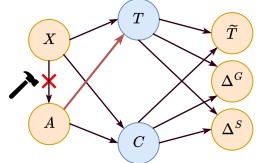

**Data:** We consider the standard setting for estimating HTEs from time-to-event data [8, 7]. That is, we consider a population $(X, A, T, C) \sim \mathbb{P}$, where $X \in \mathcal{X} \subseteq \mathbb{R}^p$ are observed covariates, $A \in \{0, 1\}$ is a binary treatment, $T \in \mathcal{T}$ event time of interest (e.g., death of the patient), and $C \in \mathcal{T}$ is the censoring time at which the patient drops out of the study. To ease notation, we assume a discrete-time setting $\mathcal{T} = \{0, \ldots, t_{\max}\}$ throughout the main part of this paper. However, all our findings can be easily extended to continuous time, and we provide the corresponding results in Appendix E.

Figure 1: **Causal graph for censored time-to-event data.** Yellow variables are observed while blue variables are unobserved. Intuitively, our goal is to recover the red arrow from $A$ to $T$ based on the observed variables.

The challenge of time-to-event data is that we cannot collect data from the full population $(X, A, T, C)$. Instead, we only observe a dataset $\mathcal{D} = \{(x_i, a_i, \tilde{t}_i, \delta_i^S, \delta_i^G)_{i=1}^n\}$ of size $n \in \mathbb{N}$ sampled i.i.d. from the population $Z = (X, A, \tilde{T}, \Delta^S, \Delta^G)$, where $\tilde{T} = \min\{T, C\}$, the indicator $\Delta^S = \mathbf{1}(T \leq C)$ equals one when the event of interest (e.g., death/recovery) is observed, and $\Delta^G = \mathbf{1}(T \geq C)$ equals

one in the censoring case (study dropout).[3] In other words, for every patient, we know only the time $\widetilde{T}$ and whether the patient experienced the main event or censoring. A causal graph is shown in Fig. 1.

**Key definitions:** We define the (conditional) *survival functions* $S_t(x, a) = \mathbb{P}(T > t \mid X = x, A = a)$ and $G_t(x, a) = \mathbb{P}(C > t \mid X = x, A = a)$ of the main event and of the censoring event, respectively. We further define the *hazard functions* $\lambda_t^S(x, a) = \mathbb{P}(\widetilde{T} = t, \Delta^S = 1 \mid \widetilde{T} \geq t, X = x, A = a)$ and $\lambda_t^G(x, a) = \mathbb{P}(\widetilde{T} = t, \Delta^G = 1 \mid \widetilde{T} \geq t, X = x, A = a)$ of the main event of interest and censoring, respectively. The survival hazard function $\lambda_t^S(x, a)$ denotes the probability that a patient with covariates $x$ and treatment $a$ experiences the main event (e.g., death) at time $t$, given survival up to time $t$. Analogously, the censoring hazard function denotes the probability that the same patient drops out at time $t$, given no prior dropout. Finally, we define the *propensity score* as $\pi(x) = \mathbb{P}(A = 1 \mid X = x)$, which represents the treatment assignment mechanism based on $X = x$.

**Causal estimand:** We use the potential outcomes framework [39] to formalize our causal inference problem. Let $T(a) \in \mathcal{T}$ denote the potential event time corresponding to a treatment intervention $A = a$. We are interested in the causal estimand

$$\tau_t(x) = \mathbb{P}(T(1) > t \mid X = x) - \mathbb{P}(T(0) > t \mid X = x) \tag{1}$$

for some fixed $t \in \mathcal{T}$. The estimand $\tau_t(x)$ is the difference in survival probability up to time $t$ for a patient with covariates $x$. We also provide extensions of all our results to conditional means $\bar{\tau}(x) = \mathbb{E}[T(1) - T(0) \mid X = x]$ as well as treatment-specific quantities such as $\mu_t(x, a) = \mathbb{P}(T(a) > t \mid X = x)$ and $\bar{\mu}(x, a) = \mathbb{E}[T(a) \mid X = x]$ in Appendix F.

**Identifiability:** We impose the following assumptions to ensure the identifiability of $\tau_t(x)$.

**Assumption 3.1** (Standard causal inference assumptions). For all $a \in \{0, 1\}$ and $x \in \mathcal{X}$ it holds: (i) *consistency*: $T(a) = T$ whenever $A = a$; (ii) *treatment overlap*: $0 < \pi(x) < 1$ whenever $\mathbb{P}(X = x) > 0$; and (iii) *ignorability*: $A \perp T(1), T(0) \mid X = x$.

**Assumption 3.2** (Survival-specific assumptions). For all $a \in \{0, 1\}$ and $x \in \mathcal{X}$ with $\mathbb{P}(X = x, A = a) > 0$ it holds: (i) *censoring overlap*: $G_{t-1}(x, a) > 0$; (ii) *survival overlap*: $S_{t-1}(x, a) > 0$; and (iii) *non-informative censoring*: $T \perp C \mid X = x, A = a$.

Assumption 3.1 is standard in causal inference [19, 36, 48] and ensures that there is (i) no interference between individuals, (ii) we have sufficient observed treatments for all covariate values, and (iii) there are no unobserved confounders that can bias our estimation. Assumption 3.2 is commonly imposed for survival analysis [4, 50] and ensures that we have sufficient non-censored and surviving individuals for each covariate value and that the censoring mechanism is independent of a patient's survival time (given covariates and treatments). Under Assumptions 3.1 and 3.2, we can identify $\tau_t(x)$ via

$$\tau_t(x) = S_t(x, 1) - S_t(x, 0) = \prod_{i=0}^{t}(1 - \lambda_i^S(x, 1)) - \prod_{i=0}^{t}(1 - \lambda_i^S(x, 0)) \tag{2}$$

We provide a proof in Appendix H. Note that the hazard functions $\lambda_i^S(x, a)$ only depend on the observed population $Z = (X, A, \widetilde{T}, \Delta^S, \Delta^G)$ and can therefore be estimated from the data $\mathcal{D}$.

**Challenges in survival analysis:** In classical causal inference, a major challenge is covariate shift, meaning a strong correlation of observed confounders with the treatment [42]. For example, specific patients may almost always receive treatments, while others almost never receive treatment. This leads to the problem of low overlap, i.e., an extreme propensity score $\pi(x)$, and thus a lack of data for specific patients with certain covariate values.

In survival analysis, the *censoring mechanism adds two additional sources of data scarcity*: (i) if $G_{t-1}(x, a)$ is small, certain patients have a low probability of being uncensored beyond time $t$, implying a lack of uncensored observations (*low censoring overlap*). Similarly, if $S_{t-1}(x, a)$ is small, most patients experience the main event before time $t$, leaving few data to estimate the hazard function $\lambda_t^S$ (*low survival overlap*). Existing learners for estimating HTEs from time-to-event data have not yet addressed challenges due to additional types of low overlap. In the following section, we provide a remedy to these challenges by proposing a general orthogonal learning framework that can incorporate custom weighting functions to address the different types of low overlap.

---

[3]Related works often set $\Delta^G = 1 - \Delta^S$, thus excluding ties (see Appendix G).

# 4 Background on orthogonal learning

**Why plug-in learners are problematic:** A straightforward method to obtain an estimator is the so-called *plugin-learner*. Here, we first obtain estimates of the survival hazards $\hat{\lambda}_i^S(x, a)$. We discuss methods for this in Appendix C. Then, we can obtain an estimator of our causal quantity of interest via $\hat{\tau}_t(x) = \hat{S}_t(x, 1) - \hat{S}_t(x, 0)$, where $\hat{S}_t(x, a) = \prod_{i=0}^{t}(1 - \hat{\lambda}_i^S(x, a))$. That is, the approach is to "plug-in" the estimated hazards into the identification formula from Eq. (2). However, it is well known in the literature that such plug-in approaches lead to so-called *"plug-in bias"* and, thus, *suboptimal* estimation [22]. For details, we refer to Appendix B.

**Why we develop two-stage learners:** As a remedy to plug-in bias, current state-of-the-art methods for HTE estimation are built upon *two-stage estimation*: First, so-called nuisance functions $\eta_t$ are estimated, which are components of the data-generating process that we will define later. Then, a second-stage learner is trained via

$$\hat{\tau}_t(x) = \arg\min_{g \in \mathcal{G}} \mathcal{L}(g, \hat{\eta}_t), \tag{3}$$

where $\mathcal{L}(g, \hat{\eta}_t)$ is some second-stage loss that depends on the estimated nuisance functions $\hat{\eta}_t$. Two-stage learners come with two main advantages: (i) they allow to estimate the causal estimand directly, thus increasing statistical efficiency; and (ii) they allow to choose a model class $\mathcal{G}$ of the causal estimand. For example, $\hat{\tau}_t(x)$ can be directly regularized, or interpretable models such as decision trees can be used.

**The benefit of orthogonal loss functions:** The current state-of-the-art for designing second-stage loss functions are so-called (Neyman-)orthogonal loss functions [5]. Formally, a second-stage loss $\mathcal{L}(g, \eta_t)$ is orthogonal if

$$D_{\eta_t} D_g \mathcal{L}(g, \eta_t)[\hat{g} - g, \hat{\eta}_t - \eta_t] = 0, \tag{4}$$

for any $\hat{g}$ and $\hat{\eta}_t$, where $D_{\eta_t}$ and $D_g$ denote directional derivatives [11]. Informally, orthogonality implies the gradient of the loss w.r.t. $g$ is insensitive to small estimation errors in the nuisance functions. This robustness w.r.t. nuisance errors often enables favorable theoretical properties of orthogonal learners, such as quasi-oracle convergence rates [11, 33].

In the following, we carefully derive orthogonal losses for the survival setting, which are currently missing in the literature. As a result, our learners allows us to address not only a lack of data due to treatment overlap issues but also due to low censoring overlap and survival overlap.

# 5 A general toolbox for obtaining orthogonal survival learners

In this section, we provide our general recipe and theory for constructing orthogonal survival learners that can be used for estimating HTEs from time-to-event data. We propose concrete learners that retarget for different overlap types in Sec. 6.

**Overview:** Our toolbox for orthogonal survival learning proceeds in three steps: ① We fit *nuisance models* that estimate the nuisance functions $\eta_t$. ② We select a *weighting function* and a corresponding *weighted target loss* that addresses a certain type of overlap violation. ③ We obtain an *orthogonalized version* of the weighted target loss that we use to fit an orthogonal second-stage learner. An overview of our toolbox is shown in Fig. 2. In contrast to existing work, our step ② addresses overlap violations beyond treatment overlap within our orthogonal learning framework.

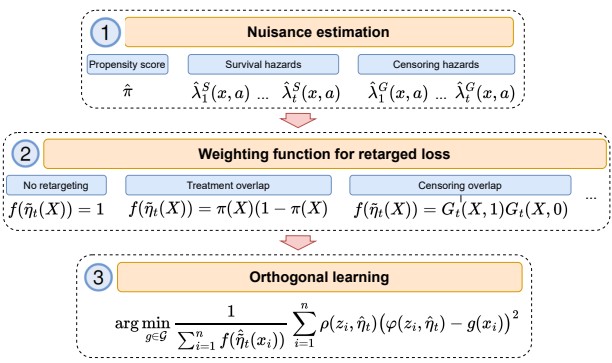

Figure 2: **Overview of the three steps of our toolbox.**

① **Nuisance estimation.** In step 1, we estimate the relevant nuisance functions for the survival setting. These include the propensity score $\pi(X)$, as well as the survival and censoring hazards

$\lambda_t^S(x, a)$ and $\lambda_t^G(x, a)$ for each time point up to $t$. Formally, we define:

$$\eta_t(X) = \left(\pi(X), \left(\lambda_i^S(X, 1), \lambda_i^S(X, 0), \lambda_i^G(X, 1), \lambda_i^G(X, 0)\right)_{i=0}^t\right) \tag{5}$$

Each of these nuisance functions can be estimated from the data using arbitrary machine learning methods. Details for estimating these functions are provided in Appendix C.

②  **Weighted target loss.** Once nuisance estimators $\hat{\eta}_t(X)$ are obtained, the second step is to define a target loss that incorporates a weighting function designed to address overlap violations. Specifically, we consider a positive weighting function $f(\widetilde{\eta}_t(X)) > 0$ that depends on $\widetilde{\eta}_t(X) = (\pi(X), S_{t-1}(X, 1), S_{t-1}(X, 0), G_{t-1}(X, 1), G_{t-1}(X, 0))$ (i.e., the propensity score, the survival and censoring functions at time $t - 1$). The corresponding *weighted target loss* is defined as

$$\bar{\mathcal{L}}_f(g, \eta_t) = \frac{1}{\mathbb{E}[f(\widetilde{\eta}_t(X))]} \mathbb{E}\left[f(\widetilde{\eta}_t(X))\left(\tau_t(X) - g(X)\right)^2\right]. \tag{6}$$

The weighted target loss represents the population loss that we aim to minimize. Note that the minimizer of $\bar{\mathcal{L}}_f(g, \eta_t)$ over a function class $\mathcal{G}$ coincides with our target quantity $\tau_t(X)$ whenever $\tau_t(X) \in \mathcal{G}$, i.e., whenever our second-stage model class $\mathcal{G}$ is sufficiently complex. This holds no matter what weighting function we use as long as we ensure that $f(\widetilde{\eta}_t(X)) > 0$.

**Intuition for the weighting function.** The weighting function $f(\widetilde{\eta}_t(X)) > 0$ allows us to *retarget* our loss towards a more favorable population. For example, by choosing $f(\widetilde{\eta}_t(X)) = \pi(X)(1 - \pi(X))$, we downweight samples with low treatment overlap, and thus retarget to the population with large treatment overlap. This allows us to prioritize estimation accuracy in regions with larger treatment overlap. Similar retargeting has been used in [20] for policy learning and in [32] for CATE estimation from uncensored data. Yet, to the best of our knowledge, we are the first to explicitly use retargeting for orthogonal HTE estimation from time-to-event data.

Weighting can be compared to clipping, i.e., removing observations with extreme weights from model training. Advantages of weighting over clipping include: (1) there is no need for choosing a cutoff value, which is otherwise often done heuristically, and (2) the weighting term is considered in the orthogonal objective we propose later, which makes the learner more robust to estimation errors in the weighting function. Clipping on the other hand can be insensitive to errors in, e.g., the propensity score.

*How to choose the weighting function?* The different types of overlap can be estimated from the data at hand by estimating $\pi$, $S_{t-1}$, and $G_{t-1}$. Based on the estimated overlap types practitioners are able to choose a weighting function that addresses the specific overlap challenges of their setting at hand. We discuss the specific weighting functions and, thus, corresponding learners in Sec. 6.

③ **Orthogonal second-stage loss.** In the final step 3, we obtain a Neyman-orthogonal version of our weighted target loss that we use for the second-stage regression. Here, we follow a similar approach as in [32], who derived weighted orthogonal learners for CATE estimation but from uncensored data.

*How to choose such an orthogonal loss?* First, we define a corresponding averaged weighted estimand as $\theta_{t,f} = \mathbb{E}[f(\widetilde{\eta}_t(X))\tau_t(X)]/\mathbb{E}[f(\widetilde{\eta}_t(X))]$. A natural candidate for an orthogonal loss is based on the so-called *efficient influence function* (EIF) $\phi_{t,f}(Z, \eta_t)$ of the parameter $\theta_{t,f}$ (see Appendix B for a more detailed background on EIFs). Further, a well-known result from semiparametric estimation theory states that the EIF satisfies the property $D_{\eta_t}\phi_{t,f}(Z, \eta_t)[\hat{\eta}_t - \eta_t]$, i.e., its directional derivatives w.r.t. the nuisance functions are zero [5, 47]. Hence, it remains to find a loss whose directional derivative w.r.t. $g$ equals to the EIF. By deriving the EIF of $\theta_{t,f}$, we obtain the following main result. We denote $\frac{\partial f}{\partial \pi}$ as the partial derivative of the function $f(\pi, \dots)$ with respect to its (functional) argument $\pi$, and use analogous notation for derivatives w.r.t. $G_{t-1}$ and $S_{t-1}$.

**Theorem 5.1.** *We define the (population) loss function*

$$\mathcal{L}_f(g, \eta_t) = \frac{1}{\mathbb{E}[f(\widetilde{\eta}_t(X))]} \mathbb{E}\left[\rho(Z, \eta_t)\left(\varphi(Z, \eta_t) - g(X)\right)^2\right], \tag{7}$$

*where*

$$\rho(Z, \eta_t) = f(\widetilde{\eta}_t(X)) + \frac{\partial f}{\partial \pi}(\widetilde{\eta}_t(X))(A - \pi(X)) \tag{8}$$

$$- \left(\frac{A}{\pi(X)} + \frac{1 - A}{1 - \pi(X)}\right)\left(\frac{\partial f(\widetilde{\eta}_t(X))}{\partial S_{t-1}(\cdot, A)} S_{t-1}(X, A)\xi_S(Z, \eta_{t-1}) + \frac{\partial f(\widetilde{\eta}_t(X))}{\partial G_{t-1}(\cdot, A)} G_{t-1}(X, A)\xi_G(Z, \eta_{t-1})\right)$$

*and*

$$\varphi(Z, \eta_t) = S_t(X, 1) - S_t(X, 0) - \frac{(A - \pi(X))\xi_S(Z, \eta_t)S_t(X, A)f(\widetilde{\eta}_t(X))}{\pi(X)(1 - \pi(X))\rho(Z, \eta_t)} \tag{9}$$

*with*

$$\xi_S(Z, \eta_t) = \sum_{i=0}^{t} \frac{\mathbf{1}(\widetilde{T} = i, \Delta^S = 1) - \mathbf{1}(\widetilde{T} \geq i)\lambda_i^S(X, A)}{S_i(X, A)G_{i-1}(X, A)}, \quad \xi_G(Z, \eta_{t-1}) = \sum_{i=0}^{t-1} \frac{\mathbf{1}(\widetilde{T} = i, \Delta^G = 1) - \mathbf{1}(\widetilde{T} \geq i)\lambda_i^G(X, A)}{S_{i-1}(X, A)G_i(X, A)}, \tag{10}$$

*and where we used the convention $S_{-1}(x, a) = G_{-1}(x, a) = 1$. Then, $\mathcal{L}_f(g, \eta_t)$ is orthogonal with respect to the nuisance functions $\eta_t$.*

*Proof.* See Appendix H. Therein, we calculate the functional derivatives according to Eq. (4). □

Theorem 5.1 shows that the loss $\mathcal{L}_f(g, \eta_t)$ is orthogonal for any choice of weighting function $f$ and thus robust with respect to estimation errors in the nuisance functions. Of note, orthogonality implies beneficial convergence rates, as shown in [11] for general orthogonal and convex losses. It remains to show that minimizing $\mathcal{L}_f(g, \eta_t)$ actually leads to a meaningful estimator, i.e., we actually obtained an orthogonalized version of our weighted target loss.

**Theorem 5.2.** *Let $g_f^* = \arg\min_{g \in \mathcal{G}} \mathcal{L}_f(g, \eta_t)$ be the minimizer of the orthogonal loss from Eq. (7) over a class of functions $\mathcal{G}$. Then, $g_f^*$ also minimizes the weighted target loss*

$$g_f^* = \arg\min_{g \in \mathcal{G}} \frac{1}{\mathbb{E}[f(\widetilde{\eta}_t(X))]} \mathbb{E}\left[f(\widetilde{\eta}_t(X))(\tau_t(X) - g(X))^2\right]. \tag{11}$$

*Hence, $g_f^* = \tau_t$ for any weighting function $f$ as long as $\tau_t \in \mathcal{G}$.*

*Proof.* See Appendix H. □

Theorem 5.2 implies that minimizing the orthogonal loss $\mathcal{L}_f(g, \eta_t)$ indeed leads to a consistent estimator of the causal estimand of interest no matter what weighting function $f$ we choose (assuming the model class $\mathcal{G}$ is large enough to contain the ground-truth causal estimand and the nuisance functions are estimated sufficiently well). As a result, Theorem 5.1 and Theorem 5.2 imply together that the loss $\mathcal{L}_f(g, \eta_t)$ is exactly what we wanted to derive: *an orthogonalized version of our weighted target loss*. It can be readily used as a second-stage loss for obtaining the causal target parameter by minimizing its corresponding empirical version with estimated nuisance functions from step 1, i.e.,

$$\hat{\tau}_t(x) = \arg\min_{g \in \mathcal{G}} \frac{1}{\sum_{i=1}^{n} f(\hat{\widetilde{\eta}}_t(x_i))} \sum_{i=1}^{n} \rho(z_i, \hat{\eta}_t)(\varphi(z_i, \hat{\eta}_t) - g(x_i))^2. \tag{12}$$

## 6  Orthogonal survival learners

We now explicitly instantiate our toolbox for specific weighting functions and write down the corresponding survival learners we obtain. We show that our toolbox both encompasses existing learners as special cases (Survival-DR- and Survival-R-learner), but also leads to novel learners that address overlap types specific to the survival setting. We use the letters T/C/S in typewriter font to refer to survival learners addressing specific variants of overlap. We use $\emptyset$ to refer to a learner that does not address any type of overlap.

$\emptyset$**-learner (no weighting; also known as Survival DR-learner [32, 52]):** Here, no weighting is used in the target loss from Eq. (6), i.e., $f(\widetilde{\eta}_t(X)) = 1$. As a consequence, it holds that $\rho(Z, \eta_t) = 1$, and the orthogonal loss becomes

$$\mathcal{L}_{\mathrm{DR}}(g, \eta_t) = \mathbb{E}\left[\left(S_t(X, 1) - S_t(X, 0) + \frac{(A - \pi(X))}{\pi(X)(1 - \pi(X))}(Y(\eta_t) - S_t(X, A)) - g(X)\right)^2\right] \tag{13}$$

using the transformed outcome $Y(\eta_t) = S_t(X, A)(1 - \xi_S(Z, \eta_t))$. The drawback of the (Survival)-DR-learner is that it is sensitive to *all* types of low overlap as it includes divisions by $\pi(X), 1 - \pi(X)$, $S_i(X, A)$, and $G_i(X, A)$. If one of these quantities is small, the DR-loss becomes unstable, and the learner will exhibit high variance.

T**-learner**[4] **(treatment overlap; also known as Survival R-learner) [52]:** To address treatment overlap, we choose $f(\widetilde{\eta}_t(X)) = \pi(X)(1 - \pi(X))$. In other words, individuals with small treatment overlap will be down-weighted in the weighted target loss. By noting that $\rho(Z, \eta_t) = (A - \pi(X))^2$, the orthogonal loss becomes

$$\mathcal{L}_{\text{R}}(g, \eta_t) = \mathbb{E}\left[\left(\widetilde{Y}(\eta_t) - \widetilde{A}(\eta_t)g(X)\right)^2\right], \tag{14}$$

with transformed variables $\widetilde{A}(\eta_t) = A - \pi(X)$ and $\widetilde{Y}(\eta_t) = Y(\eta_t) - S_t(X)$ with $S_t(X) = \mathbb{P}(S > t \mid X) = \pi(X)S_t(X, 1) + (1 - \pi(X))S_t(X, 0)$, as proposed in [52]. Compared to the DR-learner, the R-learner does not divide by $\pi(X)$ or $1 - \pi(X)$ and is thus less sensitive w.r.t. small or large propensities (low treatment overlap). However, the R-learner still divides by $S_t(X, A)$, and $G_t(X, A)$, and is thus sensitive w.r.t. low censoring or survival overlap.

C**-learner (censoring overlap):** To address low censoring overlap, we can choose the weighting function $f(\widetilde{\eta}_t(X)) = G_{t-1}(X, 1)G_{t-1}(X, 0)$, which down-weights patients who have a large treated or untreated censoring probability before time $t$. In other words, if, for a patient with covariates $X$, either the treated or untreated probability to remain in the study until the time $t$ of interest is small, the patient will be down-weighted in the target loss. This type of weighting results in

$$\rho(Z, \eta_t) = G_{t-1}(X, 1)G_{t-1}(X, 0)\left(1 - \left(\frac{A}{\pi(X)} + \frac{1 - A}{1 - \pi(X)}\right)\xi_G(Z, \eta_{t-1})\right), \tag{15}$$

$$\varphi(Z, \eta) = S_t(X, 1) - S_t(X, 0) - \frac{(A - \pi(X))\xi_S(Z, \eta_t)S_t(X, A)}{\pi(X)(1 - \pi(X))\left(1 - \left(\frac{A}{\pi(X)} + \frac{1 - A}{1 - \pi(X)}\right)\xi_G(Z, \eta_{t-1})\right)}.$$

Both the multiplication by $G_{t-1}(X, 1)G_{t-1}(X, 0)$ in Eq. (15) and the division by $\xi_G(Z, \eta_{t-1})$ below downweight possible extreme loss values induced by low censoring overlap via division by $G_i(X, A)$ in $\xi_G(Z, \eta_{t-1})$. However, in contrast to the R-learner, divisions by propensities $\pi(X)$ still occur in the loss (sensitivity to treatment overlap).

S**-learner (survival overlap):** Analogously to censoring overlap, we can also weight for survival overlap via $f(\widetilde{\eta}_t(X)) = S_{t-1}(X, 1)S_{t-1}(X, 0)$. That is, we down-weight patients who have a low treated or untreated survival probability beyond time $t - 1$. This results in weighting results in

$$\rho(Z, \eta_t) = S_{t-1}(X, 1)S_{t-1}(X, 0)\left(1 - \left(\frac{A}{\pi(X)} + \frac{1 - A}{1 - \pi(X)}\right)\xi_S(Z, \eta_{t-1})\right), \tag{16}$$

$$\varphi(Z, \eta) = S_t(X, 1) - S_t(X, 0) - \frac{(A - \pi(X))\xi_S(Z, \eta_t)S_t(X, A)}{\pi(X)(1 - \pi(X))\left(1 - \left(\frac{A}{\pi(X)} + \frac{1 - A}{1 - \pi(X)}\right)\xi_S(Z, \eta_{t-1})\right)},$$

which is less sensitive to divisions by $S_i(X, A)$ as compared to the DR-learner loss.

**Combined overlap types:** It is also possible to arbitrarily combine weighting to accommodate different overlap types simultaneously. This results in the following learners:

- T+C**-learner (treatment-censoring overlap):** $f(\widetilde{\eta}_t(X)) = \pi(X)(1 - \pi(X))G_{t-1}(X, 1)G_{t-1}(X, 0)$;
- T+S**-learner (treatment-survival overlap):** $f(\widetilde{\eta}_t(X)) = \pi(X)(1 - \pi(X))S_{t-1}(X, 1)S_{t-1}(X, 0)$;
- C+S**-learner (censoring-survival overlap):** $f(\widetilde{\eta}_t(X)) = S_{t-1}(X, 1)S_{t-1}(X, 0)G_{t-1}(X, 1)G_{t-1}(X, 0)$;
- T+C+S**-learner (all):** $f(\widetilde{\eta}_t(X)) = \pi(X)(1 - \pi(X))S_{t-1}(X, 1)S_{t-1}(X, 0)G_{t-1}(X, 1)G_{t-1}(X, 0)$.

**Choice of learner/ weighting function.** The choice of weighting function depends on the type(s) of overlap we would like our learner to be robust for (similar to choosing between DR- and R-learner in standard causal inference). In practice, we recommend using the estimated nuisance functions to inspect overlap (e.g., by visualizing the propensity score, censoring, and survival functions).

## 7 Experiments

We follow best-practice in causal inference literature (e.g., [8, 13]) and perform experiments using synthetic and real-world data to demonstrate the effectiveness of our toolbox to different types

---

[4]We denote the Survival-R-learner as T-learner to make the connection to the different weighting schemes explicit.

| | No violation | Propensity | Censoring | Survival |
|---|---|---|---|---|
| ∅ | 1.86 ± 0.75 | 4.56 ± 2.75 | 3.41 ± 2.54 | 2.70 ± 1.63 |
| T | 1.86 ± 0.72 | 3.94 ± 1.90 | 3.37 ± 2.52 | 2.66 ± 1.59 |
| C | 1.93 ± 0.82 | 5.02 ± 2.99 | 1.91 ± 0.64 | 3.07 ± 1.70 |
| S | 1.99 ± 0.85 | 5.11 ± 3.09 | 2.92 ± 2.33 | 2.72 ± 1.60 |
| T+C | 1.95 ± 0.82 | 4.19 ± 2.20 | 1.87 ± 0.60 | 3.02 ± 1.67 |
| T+S | 2.02 ± 0.85 | 4.30 ± 2.28 | 2.97 ± 2.40 | 2.74 ± 1.56 |
| C+S | 2.10 ± 0.90 | 5.91 ± 3.58 | 1.90 ± 0.76 | 2.83 ± 1.63 |
| T+C+S | 2.01 ± 0.88 | 4.65 ± 2.41 | 1.86 ± 0.58 | 2.73 ± 1.45 |

| | No violation | Propensity | Censoring | Survival |
|---|---|---|---|---|
| ∅ | 1.64 ± 0.19 | 1.01 ± 0.31 | 0.61 ± 0.15 | 4.54 ± 0.36 |
| T | 1.12 ± 0.41 | 0.65 ± 0.18 | 0.75 ± 0.23 | 6.77 ± 1.08 |
| C | 1.91 ± 0.46 | 0.98 ± 0.27 | 0.60 ± 0.14 | 4.82 ± 0.46 |
| S | 2.16 ± 0.65 | 0.87 ± 0.34 | 0.60 ± 0.21 | 4.56 ± 0.70 |
| T+C | 1.40 ± 0.29 | 0.65 ± 0.18 | 0.74 ± 0.23 | 4.52 ± 0.60 |
| T+S | 3.55 ± 1.13 | 0.56 ± 0.14 | 0.71 ± 0.19 | 9.23 ± 1.27 |
| C+S | 2.71 ± 0.70 | 0.86 ± 0.31 | 0.57 ± 0.18 | 5.38 ± 0.57 |
| T+C+S | 1.35 ± 0.32 | 0.56 ± 0.13 | 0.70 ± 0.18 | 4.55 ± 0.69 |

Table 2: **PEHE in Scenario 1:** Mean and standard deviation of PEHE averaged over the first time steps across 10 runs. Targeted learners per setting (column) in gray background. ⇒ Overall, targeted weighting improves performance.

Table 3: **PEHE in Scenario 2:** Mean and standard deviation over all assessed time steps across 10 runs. Targeted learners per setting in gray background. ⇒ Again, targeted weighting generally improves performance.

of overlap violations. We instantiate all models with the *same* neural network architectures and hyperparameters. This allows us to assess the effect of our proposed weighting scheme, as differences in performance can be merely attributed to the different orthogonal loss functions for training the second-stage model. Implementation details are in Appendix I.

**Synthetic data.** *Data generation:* We consider two different data generation mechanisms: • *Scenario 1* considers a one-dimensional confounder and sigmoid propensity and hazard functions across five time steps. • *Scenario 2* follows [8] by generating 10-dimensional multivariate normal confounders with correlations across 30 time steps. From each scenario, we generate multiple different datasets, in which we introduce propensity, censoring, or survival overlap violations or a combination of them. All datasets consist of 30,000 samples. For details, see Appendix I.

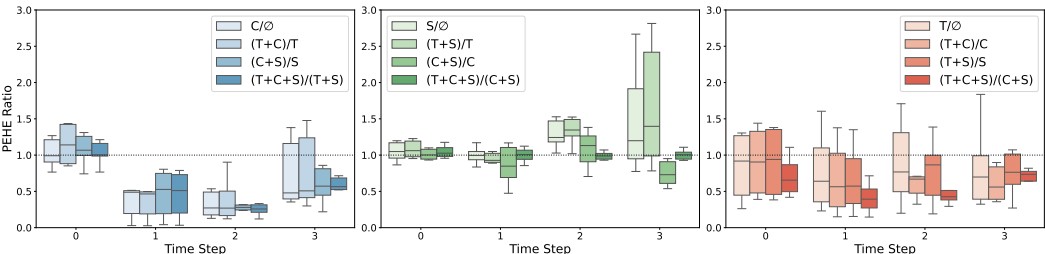

Figure 3: **Benefit of targeted weighting over time.** Ratios of PEHE of the targeted learner wrt. the learner without the correct target (data scenario I across 10 runs). **Blue:** Low censoring overlap scenario. **Green:** Low survival overlap scenario. **Red:** Low treatment overlap scenario.

*Results:* We evaluate the performance of our various orthogonal learners based on the *precision in the estimation of heterogeneous effects* (PEHE) with regard to the true CATE [17]. We compare the PEHE for scenario 1 (Table 2) and scenario (Table 3) across different types of overlap violations (i.e., no, treatment overlap, censoring overlap, and survival overlap violation, respectively). For better comparability, we report the PEHE $\times 10^{-4}$. Across both scenarios, we observe that *the learners targeted for the low-overlap scenario achieve the lowest PEHE*. Furthermore, targeted weighting reduces the estimation variance but unsuitable weighting can harm performance.

In Figure 3, we further show the benefit of our targeted weighted learners in terms of PEHE ratios at different time steps. For data scenarios with low censoring or survival overlap, we observe a decreasing benefit of our learners over time after an initially equal performance of all learners. This is in line with our expectations: (i) At timestep 0, no censoring or time-dependent survival hazards are present. Thus, the respective survival- and censoring-overlap weighting does not affect the prediction performance. (ii) The benefit of the targeted weighting reduces with increasing timesteps due to increasing hazards, decreasing sample size with $\tilde{T} \geq t$, increasing hazards, and thus decreasing effect of $f(\tilde{\eta}_t(X))$. (iii) Treatment overlap is independent of $t$. Therefore, the benefit of learners targeted to low treatment overlap is constant over time.

**Medical data.** *Data:* We perform a case study on the Twins dataset as in [31] to showcase the applicability of our learners to high-dimensional medical data. The dataset considers the birth weight of 11984 pairs of twins born in the USA between 1989 and 1991 with respect to mortality in the first year of life. Treatment $a = 1$ corresponds to being born the heavier twin. The dataset contains 46

confounders. We provide more information on the data and preprocessing in Supplement I. For a detailed description of the dataset, see [31].

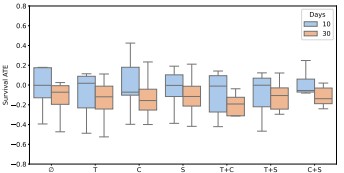

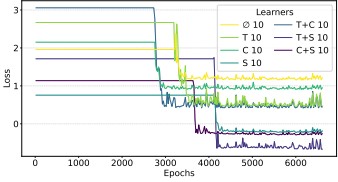

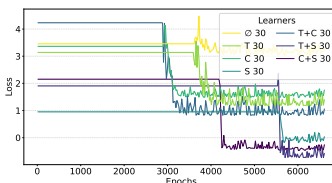

Figure 4: **Twins:** 10- and 30-day effects across 10 runs ⇒ Estimated survival effects align with the literature. Variance decreases for weighted learners.

Figure 5: **Validation loss for 10-day survival:** Fastest convergence for the C- and T+C-learners, indicating the presence of censoring and treatment overlap violations.

Figure 6: **Validation loss for 30-day survival:** As in Fig.5, the C and T+C learners converge fastest, reproducing the former finding.

*Results:* We analyze the effect on survival after 10 days and one month. We observe generally a negative effect on mortality, i.e., a positive effect on survival, for being born heavier (see Fig. 4). This is in line with the literature and medical results [e.g., 31]. We also observe an increasing effect over time, i.e., from 10 to 30 days, which again is in line with medical domain knowledge. Overall, we observe a lower estimation variance for the T+C (for treatment-censoring overlap) and the C+S-learner learners (for censoring-survival overlap). This suggests the presence of multiple forms of overlap violations, especially censoring overlap, in the data, and underlines the necessity of our survival learners targeted for specific overlap types.

The benefits of our proposed orthogonal survival learners show when inspecting the validation loss during training (Fig. 5 and 6): The C- and C+T-learners show by far the fastest convergence, whereas survival-overlap weighting in the S-learner hinders fast convergence. This affirms that suitably weighted learners are close to the data-generating process, enabling fast convergence, which brings important benefits for estimation in fine-sample regimes as common in medical applications. We note that high learning rates are likely to result in oscillating behavior on the loss, as the reweighting can be initially unstable to optimize. Therefore, we follow best practice and employ a low learning rate together with the optimizer [25], which adapts learning rates over time and may switch from small to larger gradients, which can thus result in larger drops in the validation loss.

## 8  Discussion

**Limitations.** We observed that appropriately targeted weighted learners achieve better estimation performance in terms of PEHE and improve convergence speed. However, in practice, we recommend carefully assessing the necessary weighting before applying our toolbox, as inappropriate weighting, i.e., overlap weighting even if there is sufficient overlap, can significantly slow down convergence. We recommended plotting the estimated weighting function (e.g., propensity or censoring overlap) as a measure of "trustworthiness" (or uncertainty) of the model predictions. Predictions in low-overlap regions may then be discarded or deferred to domain experts, while the model predictions in large-overlap regions benefit from weighting and may be leveraged. Finally, we note that *complete* overlap violations can still lead to unstable training and result in a high variance of the estimate, as is common with weighted orthogonal learners.

**Broader impact.** Our toolbox has a crucial impact on HTE estimation in personalized medicine. Time-to-event data is common in medical settings, but frequently suffers from censoring-induced censoring and survival overlap violations. Our toolbox offers a way to ensure reliable and stable treatment effect estimation in such settings.

**Conclusion.** We proposed a toolbox for constructing custom-weighted orthogonal survival learners to estimate HTEs from time-to-event data. Our learners can be constructed in a model-agnostic way, are semi-parametrically efficient, and ensure stable training in the presence of treatment, censoring, or survival overlap violations. As a result, our work makes an important step towards reliable estimation of heterogeneous treatment effects in survival settings.

## Acknowledgements

We thank Lars van der Laan and Jonas Schweisthal for helpful discussions.

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

# A  Extended related work

**Semiparametric inference and orthogonal learning:** The concept of Neyman orthogonality is deeply rooted in semiparametric efficiency theory [22, 49]. Neyman-orthogonal and efficient-influence function-based estimators have a long tradition in causal inference, primarily for the estimation of average treatment effects. Examples include the AIPTW estimator [37], TMLE [48], the doubleML framework [5], and doubly robust policy value estimation [2]. Recently, the concept of Neyman orthogonality has been extended to HTEs [11], which allowed the construction of various orthogonal learners, including the DR- and R-learner for conditional average treatment effects [23, 46, 32, 33].

**Efficient average treatment effect estimation for time-to-event data:** Most work on semiparametric inference for time-to-event data has focused on *average treatment effects* (ATEs). For example, [38] proposed so-called doubly robust censoring unbiased transformations for semiparametric efficient inference on the ATE under censoring. Furthermore, various works proposed one-step Targeted Maximum Likelihood Estimators (TMLE) for estimating average causal quantities in survival settings [4, 50, 56].

# B    Background on influence functions and orthogonal learning

In the following, we provide a short background on efficient influence functions and orthogonal learning. We mostly follow Kennedy [22].

**Efficient influence function (EIF).** In statistics, estimation is formalized via statistical model $\{\mathbb{P} \in \mathcal{P}\}$, where $\mathcal{P}$ is a family of probability distributions. We are interested in estimating a functional $\psi \colon \mathcal{P} \to \mathbb{R}$. For example, $\psi(\mathbb{P}) = \mathbb{E}[S_t(X, 1) - S_t(X, 0)] = \mathbb{E}[\tau_t(X)]$. If $\psi$ is sufficiently smooth, it admits the so-called *von Mises* or *distributional Taylor expansion*

$$\psi(\bar{\mathbb{P}}) - \psi(\mathbb{P}) = \int \phi(t, \bar{\mathbb{P}}) \, \mathrm{d}(\bar{\mathbb{P}} - \mathbb{P})(t) + R_2(\bar{\mathbb{P}}, \mathbb{P}), \tag{17}$$

where $R_2(\bar{\mathbb{P}}, \mathbb{P})$ is a *second-order remainder term* and $\phi(t, \mathbb{P})$ is the so-called *efficient influence function* of $\psi$, satisfying $\int \phi(t, \mathbb{P}) d\mathbb{P}(t) = 0$ and $\int \phi(t, \mathbb{P})^2 d\mathbb{P}(t) < \infty$.

**Plug-in bias and debiased inference.** Let now $\hat{\mathbb{P}}$ be an estimator of $\mathbb{P}$ and $\psi(\hat{\mathbb{P}})$ the so-called *plug-in estimator* of $\psi(\mathbb{P})$. The von Mises expansion from Eq. (17) implies that $\psi(\hat{\mathbb{P}})$ yields a first-order *plug-in bias* because

$$\psi(\hat{\mathbb{P}}) - \psi(\mathbb{P}) = -\int \phi(t, \hat{\mathbb{P}}) \, \mathrm{d}\mathbb{P}(t) + R_2(\hat{\mathbb{P}}, \mathbb{P}) \tag{18}$$

due to that $\int \phi(t, \hat{\mathbb{P}}) \, \mathrm{d}\hat{\mathbb{P}}(t) = 0$. In other words, simply plugging the estimated nuisance functions into the identification formula can result in a biased estimator.

A simple way to correct for the plug-in bias is to estimate the bias term from the right-hand side of Eq. (18) and add it to the plug-in estimator via

$$\hat{\psi}^{\text{A-IPTW}} = \psi(\hat{\mathbb{P}}) + \mathbb{P}_n(\phi(T, \hat{\mathbb{P}})). \tag{19}$$

this estimator is called (one-step) bias-corrected estimator.

**Debiased target loss.** One-step bias correction generally only works for finite-dimensional target quantities (e.g., average causal effects such as $\mathbb{E}[\tau_t(X)]$). In this paper, however, we are interested in the HTE $\tau_t(X)$, which is an infinite-dimensional target quantity. Hence, direct one-step bias correction is not applicable. The EIF can nevertheless be used for obtaining a "good" estimator of the HTE. The idea is that, instead of de-biasing the HTE $\tau_t(X)$, we can de-bias the *target loss* that we aim at minimizing. This leads to orthogonal versions of the target losses, which is precisely what we derive in our paper.

## C Estimation of nuisance functions

Here, we discuss the estimation of the nuisance functions, that is, the propensity score $\pi(x)$, and the hazard functions $\lambda_j^S(x, a)$ and $\lambda_j^G(x, a)$.

**Propensity score.** The propensity score $\mathbb{P}(A = 1 \mid X = x)$ defines a standard binary classification problem. Hence, any standard classification algorithm (such as a feed-forward neural network with softmax activation and cross-entropy loss) can be used for propensity estimation.

**Hazard functions.** We can estimate the hazard functions via *maximum likelihood*. We can write the (population) likelihood as

$$\mathbb{P}(\widetilde{T} = \widetilde{t}, \Delta^S = \delta^S, \Delta^G = \delta^G \mid X = x, A = a) \tag{20}$$

$$= \prod_{j=0}^{\widetilde{t}-\delta^S} \left(1 - \lambda_j^S(x, a)\right) \lambda_{\widetilde{t}}^S(x, a)^{\delta^S} g_{\widetilde{t}}(x, a)^{\delta^S \delta^G + 1 - \delta^S} G_{\widetilde{t}}(x, a)^{\delta^S(1 - \delta^G)}, \tag{21}$$

where $g_{\widetilde{t}}(x, a)$ denotes the (conditional) probability mass function of $C$. Hence, we can use parametric models $\lambda_j^S(x_i, a_i, \theta)$ parametrized by $\theta$ (e.g., neural networks) to minimize the resulting the resulting log-likelihood loss

$$\mathcal{L}^S(\theta) = \sum_{i=1}^{n} \sum_{j=0}^{\widetilde{t}_i - \delta_i^S} \log \left(1 - \lambda_j^S(x_i, a_i, \theta)\right) + \delta_i^S \log \left(\lambda_{\widetilde{t}_i}^S(x_i, a_i, \theta)\right). \tag{22}$$

Analogously, $\lambda_j^G(x_i, a_i, \theta)$ can be trained to minimize

$$\mathcal{L}^G(\theta) = \sum_{i=1}^{n} \sum_{j=0}^{\widetilde{t}_i - \delta_i^G} \log \left(1 - \lambda_j^G(x_i, a_i, \theta)\right) + \delta_i^G \log \left(\lambda_{\widetilde{t}_i}^G(x_i, a_i, \theta)\right). \tag{23}$$

## D  Extensions to marginalized effects

We now discuss an extension to marginalized effects, i.e., the case where we are interested in the causal estimand

$$\tau_t(v) = \mathbb{P}(T(1) > t \mid V = v) - \mathbb{P}(T(0) > t \mid V = v) \tag{24}$$

conditioned on a subset of confounders $V \subset X$. This can be relevant in many applications where certain confounders are not available during inference time (called runtime confounding [6]) or should not be used (due to, e.g., fairness or privacy constraints). Under Assumptions 3.1 and 3.2, identification holds via

$$\tau_t(v) = \mathbb{E}[\tau_t(X) \mid V = v]. \tag{25}$$

For our framework in the marginalized case, we consider weighting function $f(\widetilde{\eta}_t(V))$, where the weighting only depends on the marginalized propensity score $\widetilde{\eta}_t(V) = (\pi(V) = P(A = 1 \mid V))$.

The orthogonal loss is given by

$$\mathcal{L}_f(g, \eta_t) = \frac{1}{\mathbb{E}[f(\widetilde{\eta}_t(V))]} \mathbb{E}\left[\rho(Z, \eta_t)\left(\varphi(Z, \eta_t) - g(V)\right)^2\right], \tag{26}$$

where $\rho(Z, \eta_t)$ is the same as in Eq. (7), and

$$\rho(Z, \eta_t) = f(\widetilde{\eta}_t(V)) + \frac{\partial f}{\partial \pi}(\widetilde{\eta}_t(V))(A - \pi(V)). \tag{27}$$

The following theorem states that this actually targets a meaningful weighted loss.

**Theorem D.1.** *Let* $g_f^* = \arg\min_{g \in \mathcal{G}} \mathcal{L}_f(g, \eta_t)$ *be the minimizer of the orthogonal loss from Eq. (26) over a class of functions* $\mathcal{G}$. *Then,* $g_f^*$ *also minimizes the oracle loss*

$$g_f^* = \arg\min_{g \in \mathcal{G}} \frac{1}{\mathbb{E}[f(\widetilde{\eta}_t(V))]} \mathbb{E}\left[f(\widetilde{\eta}_t(V))\left(\tau_t(V) - g(V)\right)^2\right] \tag{28}$$

*Hence,* $g_f^* = \tau_t$ *for any weighting function* $f$ *as long as* $\tau_t \in \mathcal{G}$.

*Proof.* We write the orthogonal loss as

$$\mathcal{L}_f(g, \eta_t) = \frac{1}{\mathbb{E}[f(\widetilde{\eta}_t(V))]} \mathbb{E}\left[\rho(Z, \eta_t)\left(\varphi(Z, \eta_t) - g(V)\right)^2\right] \tag{29}$$

$$= \frac{1}{\mathbb{E}[f(\widetilde{\eta}_t(V))]} \mathbb{E}\left[\rho(Z, \eta_t)\left(\varphi(Z, \eta_t) - \tau_t(V) + \tau_t(V) - g(V)\right)^2\right] \tag{30}$$

$$= \frac{1}{\mathbb{E}[f(\widetilde{\eta}_t(V))]} \left(\mathbb{E}\left[\rho(Z, \eta_t)\left(\varphi(Z, \eta_t) - \tau_t(V)\right)^2\right]\right. \tag{31}$$

$$\left. + 2\mathbb{E}\left[\rho(Z, \eta_t)\left(\varphi(Z, \eta_t) - \tau_t(V)\right)\tau_t(V)\right]\right) \tag{32}$$

$$+ \frac{1}{\mathbb{E}[f(\widetilde{\eta}_t(V))]}\left(-2\mathbb{E}\left[\rho(Z, \eta_t)\left(\varphi(Z, \eta_t) - \tau_t(V)\right)g(V)\right]\right. \tag{33}$$

$$\left. + \mathbb{E}\left[\rho(Z, \eta_t)\left(\tau_t(V) - g(V)\right)^2\right]\right), \tag{34}$$

where only the terms in the last equation depend on $g$. The first term we can rewrite as

$$\mathbb{E}\left[\rho(Z, \eta_t)\left(\varphi(Z, \eta_t) - \tau_t(V)\right)g(V)\right] \tag{35}$$

$$= \mathbb{E}\left[\rho(Z, \eta_t)\left(S_t(X, 1) - S_t(X, 0) - \frac{(A - \pi(X))\xi_S(Z, \eta_t)S_t(X, A)f(\widetilde{\eta}_t(X))}{\pi(X)(1 - \pi(X))\rho(Z, \eta_t)} - \tau_t(V)\right)g(V)\right] \tag{36}$$

$$= \mathbb{E}\left[\mathbb{E}\left[\rho(Z, \eta_t)\left(S_t(X, 1) - S_t(X, 0) - \tau_t(V)\right)g(V)\Big|V, A\right]\right] \tag{37}$$

$$- \mathbb{E}\left[\mathbb{E}\left[\frac{(A - \pi(X))\xi_S(Z, \eta_t)S_t(X, A)f(\widetilde{\eta}_t(X))}{\pi(X)(1 - \pi(X))}g(V)\Big|X, A\right]\right] \tag{38}$$

$$= \mathbb{E}\left[\rho(Z, \eta_t)g(V)\left(\mathbb{E}\left[S_t(X, 1) - S_t(X, 0)\Big|V\right] - \tau_t(V)\right)\right] \tag{39}$$

$$- \mathbb{E}\left[\frac{(A - \pi(X))\mathbb{E}[\xi_S(Z, \eta_t) \mid X, A]S_t(X, A)f(\widetilde{\eta}_t(X))}{\pi(X)(1 - \pi(X))}g(V)\right] \tag{40}$$

$$\overset{(*)}{=} 0, \tag{41}$$

where $(*)$ follows from Lemma H.2. For the second term, note that

$$\mathbb{E}\left[\rho(Z,\eta_t) \mid V\right] \tag{42}$$

$$=\mathbb{E}\left[f(\widetilde{\eta}_t(V)) + \frac{\partial f}{\partial \pi}(\widetilde{\eta}_t(V))(A - \pi(V))\right. \tag{43}$$

$$\left.- \bigg| V\right] \tag{44}$$

$$=f(\widetilde{\eta}_t(V)) + \frac{\partial f}{\partial \pi}(\widetilde{\eta}_t(V))(\pi(V) - \pi(V)) \tag{45}$$

$$\overset{(**)}{=} f(\widetilde{\eta}_t(V)), \tag{46}$$

where $(**)$ follows from Lemma H.2. Hence,

$$\mathbb{E}\left[\rho(Z,\eta_t)\left(\tau_t(V) - g(V)\right)^2\right] = \mathbb{E}\left[\mathbb{E}\left[\rho(Z,\eta_t)\left(\tau_t(V) - g(V)\right)^2 \mid V\right]\right] \tag{47}$$

$$= \mathbb{E}\left[\mathbb{E}\left[f(\widetilde{\eta}_t(V))\left(\tau_t(V) - g(V)\right)^2 \mid V\right]\right] \tag{48}$$

$$= \mathbb{E}\left[f(\widetilde{\eta}_t(V))\left(\tau_t(V) - g(V)\right)^2\right]. \tag{49}$$

Putting everything together, we obtain

$$\mathcal{L}_f(g,\eta_t) = \frac{1}{\mathbb{E}[f(\widetilde{\eta}_t(V))]}\left(\mathbb{E}\left[\rho(Z,\eta_t)\left(\varphi(Z,\eta_t) - \tau_t(V)\right)^2\right]\right. \tag{50}$$

$$+2\mathbb{E}\left[\rho(Z,\eta_t)\left(\varphi(Z,\eta_t) - \tau_t(V)\right)\tau_t(V)\right]) \tag{51}$$

$$+ \frac{1}{\mathbb{E}[f(\widetilde{\eta}_t(V))]}\mathbb{E}\left[f(\widetilde{\eta}_t(V))\left(\tau_t(V) - g(V)\right)^2\right], \tag{52}$$

which proves the claim because the first two summands do not depend on $g$ and do not affect the minimization. $\qquad\square$

As a consequence, we obtain the following two learners (which are, to the best of our knowledge, novel).

**Marginalized survival DR-learner (no weighting; in our taxonomy: marginalized $\emptyset$-learner).** Here, we set $f(\widetilde{\eta}_t(V)) = 1$, which implies $\rho(Z,\eta_t) = 1$ and

$$\mathcal{L}_{DR}(g,\eta_t) = \mathbb{E}\left[\left(\varphi(Z,\eta_t) - g(V)\right)^2\right] \tag{53}$$

with

$$\varphi(Z,\eta) = S_t(X,1) - S_t(X,0) - \frac{(A - \pi(X))\xi_S(Z,\eta_t)S_t(X,A)}{\pi(X)(1 - \pi(X))}. \tag{54}$$

Equivalently, this can be written as a standard DR-learner

$$\varphi(Z,\eta) = S_t(X,1) - S_t(X,0) + \frac{(A - \pi(X))}{\pi(X)(1 - \pi(X))}(Y(\eta_t) - S_t(X,A)) \tag{55}$$

with the transformed outcome $Y(\eta_t) = S_t(X,A)(1 - \xi_S(Z,\eta_t))$.

**Marginalized survival R-learner (marginalized treatment overlap; in our taxonomy: marginalized T-learner).** Here, we set $f(\widetilde{\eta}_t(X)) = \pi(V)(1 - \pi(V))$, which implies $\rho(Z,\eta_t) = (A - \pi(V))^2$. The orthogonal loss becomes

$$\mathcal{L}_R(g,\eta_t) = \mathbb{E}\left[\frac{(A - \pi(V))^2}{\mathbb{E}[\pi(V)(1 - \pi(V))]}\left(\varphi(Z,\eta_t) - g(V)\right)^2\right], \tag{56}$$

where

$$\varphi(Z,\eta) = S_t(X,1) - S_t(X,0) - w(X)\xi_S(Z,\eta_t)S_t(X,A) \tag{57}$$

with

$$w(X) = \frac{\pi(V)(1 - \pi(V))}{\pi(X)(1 - \pi(X))}\frac{(A - \pi(X))}{(A - \pi(V))^2}. \tag{58}$$

This is equivalent to minimizing the R-learner loss

$$\mathcal{L}_R(g, \eta_t) = \mathbb{E}\left[\left(\widetilde{Y}(\eta_t) - \widetilde{A}(\widetilde{\eta}_t)g(V)\right)^2\right], \tag{59}$$

where $\widetilde{A}(\widetilde{\eta}_t) = A - \pi(V)$ and

$$\widetilde{Y}(\eta_t) = w(X)Y(\eta_t) + A\left[(1 - w(X))S_t(X, 1) + (w(X) - 1)S_t(X, 0))\right] \tag{60}$$
$$- \left[\pi(V)S_t(X, 1) + (w(X) - \pi(V))S_t(X, 0)\right]. \tag{61}$$

Note that this coincides with the standard survival R-learner for $V = X$ as this implies $w(X) = 1$.

# E   Extension to the continuous-time setting

Our survival learners also extend to the continuous-time setting by making two key changes: (i) we have to estimate the hazard function in a different way, and (ii) we have to add a small modification to our weighted orthogonal loss.

**(i) Hazard functions in continuous time.** In continuous time, we can write

$$\lambda_t^S(x,a) = \mathbb{P}(\widetilde{T} = t, \Delta^S = 1 \mid \widetilde{T} \geq t, X = x, A = a) \tag{62}$$

$$= \frac{\mathbb{P}(\widetilde{T} = t, \Delta^S = 1 \mid X = x, A = a)}{\mathbb{P}(\widetilde{T} \geq t \mid X = x, A = a)} \tag{63}$$

$$= \frac{\mathbb{P}(\widetilde{T} = t \mid \Delta^S = 1, X = x, A = a)\mathbb{P}(\Delta^S = 1 \mid X = x, A = a)}{\sum_{k \geq t} \mathbb{P}(\widetilde{T} = k \mid X = x, A = a)} \tag{64}$$

$$= \frac{\mathbb{P}(\widetilde{T} = t \mid \Delta^S = 1, X = x, A = a)\mathbb{P}(\Delta^S = 1 \mid X = x, A = a)}{\sum_{k \geq t} \mathbb{P}(\widetilde{T} = k \mid X = x, A = a)} \tag{65}$$

$$\tag{66}$$

and analogously

$$\lambda_t^G(x,a) = \frac{\mathbb{P}(\widetilde{T} = t \mid \Delta^G = 1, X = x, A = a)\mathbb{P}(\Delta^G = 1 \mid X = x, A = a)}{\sum_{k \geq t} \mathbb{P}(\widetilde{T} = k \mid X = x, A = a)}. \tag{67}$$

Hence, we can estimate the hazards by estimating the conditional probabilities $\mathbb{P}(\widetilde{T} = t \mid \Delta^S = 1, X = x, A = a)$, $\mathbb{P}(\Delta^S = 1 \mid X = x, A = a)$, and $\mathbb{P}(\widetilde{T} = t \mid X = x, A = a)$ for all $t$. This can be done by using standard classification algorithms such a feed-forward neural networks with softwax activation and cross-entropy loss. As an alternative, one can impose parametric assumptions such as the Cox-model, as done in [21].

**(ii) Orthogonal loss.** For the second stage, we can use the same loss as in Eq. (7) but where we define

$$\xi_S(Z, \eta_t) = \frac{\mathbf{1}(\widetilde{T} \leq t, \Delta = 1)}{S_{\widetilde{T}}(X, A)G_{\widetilde{T}}(X, A)} - \int_0^t \frac{\mathbf{1}(\widetilde{T} \geq i)\lambda_i^S(X, A)}{S_i(X, A)G_i(X, A)} \, \mathrm{d}i \tag{68}$$

as well as

$$\xi_G(Z, \eta_t) = \frac{\mathbf{1}(\widetilde{T} \leq t, \Delta = 0)}{S_{\widetilde{T}}(X, A)G_{\widetilde{T}}(X, A)} - \int_0^t \frac{\mathbf{1}(\widetilde{T} \geq i)\lambda_i^G(X, A)}{S_i(X, A)G_i(X, A)} \, \mathrm{d}i. \tag{69}$$

Here, the integrals can be approximated using a numerical integration method.

# F Extensions to further causal estimands

## F.1 Treatment-specific survival probability

We aim to construct (weighted) orthogonal learners to estimate the survival probability

$$\mu_{a,t}(x) = \mathbb{P}(T(a) > t \mid X = x) \tag{70}$$

specific for a fixed treatment $A = a$. We consider a weighting function $f(\widetilde{\eta}_t(X))$ that depends on the treatment-specific nuisance functions $\widetilde{\eta}_t(X) = (\pi_a(X), S_{t-1}(X, a), G_{t-1}(X, a))$, where $\pi_a(x) = \mathbb{P}(A = a \mid X = x)$. Following the same derivation as in our main paper, we first define the corresponding weighted average treatment effect via

$$\theta_{a,t,f} = \frac{\mathbb{E}[f(\widetilde{\eta}_t(X))\mu_{a,t}(X)]}{\mathbb{E}[f(\widetilde{\eta}_t(X))]}. \tag{71}$$

One can show that the efficient influence function of $\theta_{a,t,f}$ is given by

$$\phi_{a,t,f}(Z, \eta_t) = \frac{\rho_a(Z, \eta_t)}{\mathbb{E}[f(\widetilde{\eta}_t(X))]}\left(\varphi_a(Z, \eta_t) - \theta_{a,t,f}\right), \tag{72}$$

where

$$\rho_a(Z, \eta_t) = f(\widetilde{\eta}_t(X)) + \frac{\partial f}{\partial \pi_a}(\widetilde{\eta}_t(X))(\mathbf{1}(A = a) - \pi_a(X)) \tag{73}$$

$$- \frac{\mathbf{1}(A = a)}{\pi_a(X)}\left(\frac{\partial f(\widetilde{\eta}_t(X))}{\partial S_{t-1}(\cdot, a)}S_{t-1}(X, a)\xi_S(Z_a, \eta_{t-1})\right. \tag{74}$$

$$\left. + \frac{\partial f(\widetilde{\eta}_t(X))}{\partial G_{t-1}(\cdot, a)}G_{t-1}(X, a)\xi_G(Z_a, \eta_{t-1})\right) \tag{75}$$

and

$$\varphi_a(Z, \eta_t) = S_t(X, a) - \frac{\mathbf{1}(A = a)\xi_S(Z_a, \eta_t)S_t(X, a)f(\widetilde{\eta}_t(X))}{\pi_a(X)\rho_a(Z, \eta_t)} \tag{76}$$

and we used the notation $Z_a = (X, a, \widetilde{T}, \Delta^S, \Delta^G)$. In particular,

$$\xi_S(Z_a, \eta_t) = \sum_{i=0}^{t} \frac{\mathbf{1}(\widetilde{T} = i, \Delta^S = 1) - \mathbf{1}(\widetilde{T} \geq i)\lambda_i^S(X, a)}{S_i(X, a)G_{i-1}(X, a)}. \tag{77}$$

The orthogonal loss is given by

$$\mathcal{L}_{f,a}(g, \eta_t) = \frac{1}{\mathbb{E}[f(\widetilde{\eta}_t(X))]}\mathbb{E}\left[\rho_a(Z, \eta_t)\left(\varphi_a(Z, \eta_t) - g(X)\right)^2\right]. \tag{78}$$

## F.2 (Restricted) mean survival times

For some $h \leq t_{\max}$, we consider

$$\bar{\mu}_h(x, a) = \mathbb{E}[T(a) \wedge h \mid X = x] = \sum_{t=0}^{t_{\max}} \mathbb{P}(T \wedge h > t \mid X = x, A = a) = \sum_{t=0}^{h} S_t(x, a) \tag{79}$$

and

$$\bar{\tau}_h(x) = \mathbb{E}[T(1) \wedge h - T(0) \wedge h \mid X = x] = \sum_{t=0}^{h} S_t(x, 1) - S_t(x, 0) = \sum_{t=0}^{h} \tau_t(x). \tag{80}$$

Our weighting function now depends on $h$ and is defined via $f(\widetilde{\eta}_h(X))$. To derive the orthogonal loss, we define the averaged causal quantity as

$$\theta_{t,f} = \frac{\mathbb{E}[f(\widetilde{\eta}_h(X))\bar{\tau}_h(X)]}{\mathbb{E}[f(\widetilde{\eta}_h(X))]} = \sum_{t=0}^{h} \frac{\mathbb{E}[f(\widetilde{\eta}_h(X))\tau_t(X)]}{\mathbb{E}[f(\widetilde{\eta}_h(X))]} \tag{81}$$

Using additivity, the efficient influence function is given by

$$\phi_{t,f}(Z, \eta_t) = \frac{\rho(Z, \eta_h)}{\mathbb{E}[f(\widetilde{\eta}_h(X))]} \left(\bar{\varphi}(Z, \eta_h) - \theta_{t,f}\right), \tag{82}$$

where

$$\rho(Z, \eta_h) = f(\widetilde{\eta}_h(X)) + \frac{\partial f}{\partial \pi}(\widetilde{\eta}_h(X))(A - \pi(X)) \tag{83}$$

$$- \left(\frac{A}{\pi(X)} + \frac{1-A}{1-\pi(X)}\right) \left(\frac{\partial f(\widetilde{\eta}_h(X))}{\partial S_{h-1}(\cdot, A)} S_{h-1}(X, A)\xi_S(Z, \eta_{h-1}) \tag{84}\right.$$

$$\left. + \frac{\partial f(\widetilde{\eta}_h(X))}{\partial G_{h-1}(\cdot, A)} G_{h-1}(X, A)\xi_G(Z, \eta_{h-1})\right) \tag{85}$$

and

$$\bar{\varphi}(Z, \eta_h) = \sum_{t=0}^{h} S_t(X, 1) - S_t(X, 0) - \frac{(A - \pi(X))\xi_S(Z, \eta_t)S_t(X, A)f(\widetilde{\eta}_h(X))}{\pi(X)(1-\pi(X))\rho(Z, \eta_h)}. \tag{86}$$

The orthogonal loss is thus given by

$$\mathcal{L}_f(g, \eta_h) = \frac{1}{\mathbb{E}[f(\widetilde{\eta}_h(X))]} \mathbb{E}\left[\rho(Z, \eta_h)\left(\bar{\varphi}(Z, \eta_h) - g(X)\right)^2\right]. \tag{87}$$

# G  Extensions to unobserved ties

We assume now that $\Delta^G$ is unobserved and we only observe $\Delta = \Delta^S$. In this case, it holds that

$$\lambda_t^G(x, a) = \mathbb{P}(\widetilde{T} = t, \Delta^G = 1 \mid \widetilde{T} \geq t, X = x, A = a) \tag{88}$$

$$= \mathbb{P}(\widetilde{T} = t, \Delta = 0 \mid \widetilde{T} \geq t, X = x, A = a) + \mathbb{P}(T = t, C = t \mid \widetilde{T} \geq t, X = x, A = a) \tag{89}$$

and thus the censoring hazard is not identified from observational data as it depends on the unobserved $T$ and $C$ through $\mathbb{P}(T = t, C = t \mid \widetilde{T} \geq t, X = x, A = a)$. In the following, we proposed two methods that still allow us to apply our toolbox in this setting.

## Method 1

The term $\mathbb{P}(T = t, C = t \mid \widetilde{T} \geq t, X = x, A = a)$ quantifies the conditional probability of having a tie at time $t$. Given the independent censoring assumption, we may assume this probability will be small if $\mathcal{T}$ is sufficiently large, i.e., if we observe fine-grained time steps. Then, we can approximate

$$\lambda_t^G(x, a) \approx \mathbb{P}(\widetilde{T} = t, \Delta = 0 \mid \widetilde{T} \geq t, X = x, A = a) \tag{90}$$

and we can apply our orthogonal loss from the main paper with $\Delta^G = 1 - \Delta$. In the extreme case where $\mathcal{T}$ is continuous, equality holds and we can ignore ties altogether.

## Method 2

Here, we reparametrize our orthogonal loss using identifiable nuisance functions even if $\Delta^G$ is not observed. First, we define the survival function of the observed time

$$p_t(x, a) = \mathbb{P}(\widetilde{T} > t \mid X = x, A = a) \tag{91}$$

and note that $p_t(x, a) = S_t(x, a) G_t(x, a)$ because $\widetilde{T} = \min\{T, C\}$ (independent censoring assumption). Then, we define new nuisance functions

$$\bar{\eta}_t(X) = \pi(X) \cup (\lambda_i^S(X, 1), \lambda_i^S(X, 0), p_i(X, 1), p_i(X, 0))_{i=0}^t. \tag{92}$$

Note that the correction term $\xi_S(Z, \bar{\eta}_t)$ from Eq. (10) can now be written as

$$\xi_S(Z, \bar{\eta}_t) = \sum_{i=0}^t \frac{\mathbf{1}(\widetilde{T} = i, \Delta = 1) - \mathbf{1}(\widetilde{T} \geq i)\lambda_i^S(X, A)}{p_{i-1}(X, A)(1 - \lambda_i^S(X, A))}. \tag{93}$$

We consider a weighting function $f(\widetilde{\eta}_t(X))$, where $\widetilde{\eta}_t(X) = (\pi(X), p_{t-1}(X, 1), p_{t-1}(X, 0))$. The efficient influence function of the weighted average treatment effect $\theta_{t,f}$ is given by

$$\phi_{t,f}(Z, \bar{\eta}_t) = \frac{\bar{\rho}(Z, \bar{\eta})}{\mathbb{E}[f(\widetilde{\eta}_t(X))]} (\bar{\varphi}(Z, \bar{\eta}_t) - \theta_{t,f}), \tag{94}$$

where

$$\bar{\rho}(Z, \bar{\eta}_t) = f(\widetilde{\eta}_t(X)) + \frac{\partial f}{\partial \pi}(\widetilde{\eta}_t(X))(A - \pi(X)) \tag{95}$$

$$- \left(\frac{A}{\pi(X)} + \frac{1 - A}{1 - \pi(X)}\right) \left(\frac{\partial f(\widetilde{\eta}_t(X))}{\partial p_{t-1}(\cdot, A)} \left(\mathbf{1}(\widetilde{T} \leq t) + p_{t-1}(X, A) - 1\right)\right) \tag{96}$$

and

$$\bar{\varphi}(Z, \bar{\eta}_t) = S_t(X, 1) - S_t(X, 0) - \frac{(A - \pi(X))\xi_S(Z, \bar{\eta}_t)S_t(X, A)f(\widetilde{\eta}_t(X))}{\pi(X)(1 - \pi(X))\bar{\rho}(Z, \bar{\eta}_t)}. \tag{97}$$

The orthogonal loss becomes

$$\mathcal{L}_f(g, \bar{\eta}_t) = \frac{1}{\mathbb{E}[f(\widetilde{\eta}_t(X))]} \mathbb{E}\left[\bar{\rho}(Z, \bar{\eta}_t) (\bar{\varphi}(Z, \bar{\eta}_t) - g(X))^2\right]. \tag{98}$$

The orthogonal loss in Eq. (98) requires us to estimate $p_t(x, a)$ instead of the censoring hazards $\lambda_t^G(x, a)$. Estimating $p_t(x, a)$ for all $t$ is a standard problem of estimating a discrete conditional c.d.f. For example, one could fit a multi-output MLP with softmax activation and cross-entropy loss to estimate $\mathbb{P}(\widetilde{T} = t \mid X = x, A = a)$ and then $p_t(x, a) = 1 - \sum_{i=0}^t \mathbb{P}(\widetilde{T} = t \mid X = x, A = a)$.

In the reparametrized nuisance setting, we obtain the following orthogonal survival learners:

- (i) $\emptyset$-**learner (Survival DR-learner)** corresponds to setting $f(\widetilde{\eta}_t(X)) = 1$;
- (ii) T-**learner (Survival R-learner)** corresponds to setting $f(\widetilde{\eta}_t(X)) = \pi(X)(1 - \pi(X))$,
- (iii) S+C-**learner (survival-censoring weighting)** corresponds to setting $f(\widetilde{\eta}_t(X)) = p_{t-1}(X, 1)p_{t-1}(X, 0)$; and
- (iv) T+C+S-**learner (full weighting)** corresponds to setting $f(\widetilde{\eta}_t(X)) = \pi(X)(1 - \pi(X))p_{t-1}(X, 1)p_{t-1}(X, 0)$.

# H Proofs

## H.1 Identifiability of the target estimand

**Lemma H.1.** *Under Assumptions 3.1 and 3.2, the causal estimand $\tau_t(x)$ is identified from the observational data $Z$ via*

$$\tau_t(x) = \prod_{i=0}^{t}(1 - \lambda_i^S(x,1)) - \prod_{i=0}^{t}(1 - \lambda_i^S(x,0)). \tag{99}$$

*Proof.* We show w.l.o.g. identifiability for $\mu_t(x,a) = \mathbb{P}(T(a) > t \mid X = x)$. The result for $\tau_t(x)$ follows by taking the difference. It holds that

$$\mathbb{P}(T(a) > t \mid X = x) \stackrel{\text{(i)}}{=} \mathbb{P}(T(a) > t \mid X = x, A = a) \tag{100}$$

$$\stackrel{\text{(ii)}}{=} \mathbb{P}(T > t \mid X = x, A = a) \tag{101}$$

$$= \prod_{i=0}^{t} 1 - \mathbb{P}(T = i \mid T \geq i, X = x, A = a) \tag{102}$$

$$\stackrel{\text{(iii)}}{=} \prod_{i=0}^{t} 1 - \mathbb{P}(\widetilde{T} = i, \Delta = 1 \mid \widetilde{T} \geq i, X = x, A = a) \tag{103}$$

$$= \prod_{i=0}^{t} 1 - \lambda_i^S(x,a), \tag{104}$$

where (i) follows from ignorability and treatment overlap, (ii) from consistency, and (iii) from non-informative censoring, censoring, and survival overlap.

$\square$

## H.2 Proof of Theorem 5.1

We make use of the following lemma.

**Lemma H.2.** *Let $\xi_S(Z, \eta_t)$ and $\xi_G(Z, \eta_{t-1})$ be defined as in the main paper. Then,*

$$\mathbb{E}\left[\xi_S(Z, \eta_t) \mid X, A\right] = \mathbb{E}\left[\xi_G(Z, \eta_{t-1}) \mid X, A\right] = 0. \tag{105}$$

*Proof.* We have

$$\mathbb{E}\left[\xi_S(Z, \eta_t) \mid X, A\right] = \sum_{i=0}^{t} \mathbb{E}\left[\frac{\mathbf{1}(\widetilde{T} = i, \Delta^S = 1) - \mathbf{1}(\widetilde{T} \geq i)\lambda_i^S(X,A)}{S_i(X,A)G_{i-1}(X,A)} \middle| X, A\right] \tag{106}$$

$$= \sum_{i=0}^{t} \frac{\mathbb{P}(\widetilde{T} = i, \Delta^S = 1 \mid X, A) - \mathbb{P}(\widetilde{T} \geq i \mid X, A)\lambda_i^S(X,A)}{S_i(X,A)G_{i-1}(X,A)} \tag{107}$$

$$= \sum_{i=0}^{t} \frac{\mathbb{P}(\widetilde{T} = i, \Delta^S = 1 \mid X, A) - \mathbb{P}(\widetilde{T} = i, \Delta^S = 1 \mid X, A)}{S_i(X,A)G_{i-1}(X,A)} = 0. \tag{108}$$

The result for $\xi_G(Z, \eta_{t-1})$ follows analogously. $\square$

Now we turn to the actual proof of Theorem 5.1.

***Proof of Theorem 5.1.*** By taking the first-order directional derivative [11], we obtain

$$D_g \mathcal{L}_f(g, \eta_t)[\hat{g} - g] \tag{109}$$

$$= \frac{1}{\mathbb{E}[f(\widetilde{\eta}_t(X))]} \frac{\mathrm{d}}{\mathrm{d}t}\left[\mathbb{E}\left[\rho(Z, \eta_t)\left(\varphi(Z, \eta_t) - \{g(X) + t(\hat{g}(X) - g(X))\}\right)^2\right]\right]_{t=0} \tag{110}$$

$$= \frac{-2}{\mathbb{E}[f(\widetilde{\eta}_t(X))]} \mathbb{E}\left[\rho(Z, \eta_t)\left(\varphi(Z, \eta_t) - g(X)\right)(\hat{g}(X) - g(X))\right] \tag{111}$$

$$= \frac{-2}{\mathbb{E}[f(\widetilde{\eta}_t(X))]} \mathbb{E}\left[(\hat{g}(X) - g(X))\left\{\rho(Z, \eta_t)\left(S_t(X,1) - S_t(X,0) - g(X)\right)\right.\right. \tag{112}$$

$$\left.\left. - \frac{(A - \pi(X))S_t(X,A)f(\widetilde{\eta}(X)))}{\pi(X)(1 - \pi(X))}\right\}\right] \tag{113}$$

$$= \frac{-2}{\mathbb{E}[f(\widetilde{\eta}_t(X))]} \mathbb{E}\left[(\hat{g}(X) - g(X))\left\{f(\widetilde{\eta}_t(X))\left(S_t(X,1) - S_t(X,0) - g(X)\right.\right.\right. \tag{114}$$

$$\left.\left. - \frac{A}{\pi(X)}S_t(X,1)\xi_S(Z,\eta_t) + \frac{1-A}{1-\pi(X)}S_t(X,0)\xi_S(Z,\eta_t)\right) \tag{115}$$

$$+ \left(\frac{\partial f(\widetilde{\eta}_t(X))}{\partial \pi}(A - \pi(X)) - \frac{A}{\pi(X)}\left(\frac{\partial f(\widetilde{\eta}_t(X))}{\partial S_{t-1}(\cdot,1)}S_{t-1}(X,1)\xi_S(Z,\eta_t) + \frac{\partial f(\widetilde{\eta}_t(X))}{\partial G_{t-1}(\cdot,1)}G_{t-1}(X,1)\xi_G(Z,\eta_{t-1})\right)\right. \tag{116}$$

$$\left. - \left(\frac{1-A}{1-\pi(X)}\right)\left(\frac{\partial f(\widetilde{\eta}_t(X))}{\partial S_{t-1}(\cdot,0)}S_{t-1}(X,0)\xi_S(Z,\eta_{t-1}) + \frac{\partial f(\widetilde{\eta}_t(X))}{\partial G_{t-1}(\cdot,0)}G_{t-1}(X,0)\xi_G(Z,\eta_{t-1})\right)\right) \tag{117}$$

$$\left. \left(S_t(X,1) - S_t(X,0) - g(X)\right)\right\}\right]. \tag{118}$$

To show orthogonality, we have to show that the second-order directional derivatives w.r.t. to the nuisance functions are zero. We start by computing the derivative w.r.t. the propensity score, resulting in

$$D_\pi D_g \mathcal{L}_f(g, \eta_t)[\hat{g} - g, \hat{\pi} - \pi] \tag{119}$$

$$= \frac{d}{dt}\left[D_g\mathcal{L}_f(g, \cdots, \pi + t(\hat{\pi} - \pi))[\hat{g} - g]\right]_{t=0} \tag{120}$$

$$= \frac{-2}{\mathbb{E}[f(\widetilde{\eta}_t(X))]}\mathbb{E}\left[(\hat{g}(X) - g(X))(\hat{\pi}(X) - \pi(X))\left\{f(\widetilde{\eta}(X))\left(\frac{A}{\pi(X)^2}S_t(X,1)\xi_S(Z,\eta_t)\right.\right.\right. \tag{121}$$

$$\left.\left. + \frac{1-A}{(1-\pi(X))^2}S_t(X,0)\xi_S(Z,\eta_t)\right) \tag{122}$$

$$+ \frac{\partial f(\widetilde{\eta}_t(X))}{\partial \pi}\left(S_t(X,1) - S_t(X,0) - g(X) - \frac{A}{\pi(X)}S_t(X,1)\xi_S(Z,\eta_t) + \frac{1-A}{1-\pi(X)}S_t(X,0)\xi_S(Z,\eta_t)\right) \tag{123}$$

$$+ \left[\frac{\partial^2 f(\widetilde{\eta}_t(X))}{\partial^2 \pi}(A - \pi(X)) - \frac{\partial f(\widetilde{\eta}_t(X))}{\partial \pi}\right. \tag{124}$$

$$+ \frac{A}{\pi(X)^2}\left(\frac{\partial f(\widetilde{\eta}_t(X))}{\partial S_{t-1}(\cdot,1)}S_{t-1}(X,1)\xi_S(Z,\eta_t) + \frac{\partial f(\widetilde{\eta}_t(X))}{\partial G_{t-1}(\cdot,1)}G_{t-1}(X,1)\xi_G(Z,\eta_{t-1})\right) \tag{125}$$

$$- \frac{A}{\pi(X)}\left(\frac{\partial^2 f(\widetilde{\eta}_t(X))}{\partial \pi \partial S_{t-1}(\cdot,1)}S_{t-1}(X,1)\xi_S(Z,\eta_t) + \frac{\partial^2 f(\widetilde{\eta}_t(X))}{\partial \pi \partial G_{t-1}(\cdot,1)}G_{t-1}(X,1)\xi_G(Z,\eta_{t-1})\right) \tag{126}$$

$$- \frac{1-A}{(1-\pi(X))^2}\left(\frac{\partial f(\widetilde{\eta}_t(X))}{\partial S_{t-1}(\cdot,1)}S_{t-1}(X,1)\xi_S(Z,\eta_t) + \frac{\partial f(\widetilde{\eta}_t(X))}{\partial G_{t-1}(\cdot,1)}G_{t-1}(X,1)\xi_G(Z,\eta_{t-1})\right) \tag{127}$$

$$\left. - \left(\frac{1-A}{1-\pi(X)}\right)\left(\frac{\partial^2 f(\widetilde{\eta}_t(X))}{\partial \pi \partial S_{t-1}(\cdot,0)}S_t(X,0)\xi_S(Z,\eta_t) + \frac{\partial^2 f(\widetilde{\eta}_t(X))}{\partial \pi \partial G_{t-1}(\cdot,0)}G_{t-1}(X,0)\xi_G(Z,\eta_{t-1})\right)\right] \tag{128}$$

$$\left. \left(S_t(X,1) - S_t(X,0) - g(X)\right)\right\}\right] \tag{129}$$

$$+ \frac{2D_g\mathcal{L}_f(g, \eta_t)[\hat{g} - g]}{\mathbb{E}[f(\widetilde{\eta}_t(X))]^2}\mathbb{E}\left[\frac{\partial f(\widetilde{\eta}_t(X))}{\partial \pi}\right] \tag{130}$$

$$\overset{(*)}{=} \frac{-2}{\mathbb{E}[f(\widetilde{\eta}_t(X))]}\mathbb{E}\left[(\hat{g}(X) - g(X))(\hat{\pi}(X) - \pi(X))\left\{\frac{\partial f(\widetilde{\eta}_t(X))}{\partial \pi}\left(S_t(X,1) - S_t(X,0) - g(X)\right)\right.\right. \tag{131}$$

$$\left.\left. + \left[-\frac{\partial f(\widetilde{\eta}_t(X))}{\partial \pi}\right]\left(S_t(X,1) - S_t(X,0) - g(X)\right)\right\}\right] \tag{132}$$

$$= 0, \tag{133}$$

where in $(*)$ we applied the law of total expectation and Lemma H.2 to remove all terms including any of $\xi_S(Z, \eta_t)$, $\xi_G(Z, \eta_{t-1})$, or $A - \pi(X)$ and the same argument to show that $D_g\mathcal{L}_f(g, \eta_t)[\hat{g} - g] = 0$. This shows orthogonality w.r.t. to the propensity score.

To show orthogonality w.r.t. survival and censoring hazards, note that

$$D_{\lambda_j^S(\cdot,a)}S_k(X,a)[\hat{\lambda}_j^S(\cdot,a) - \lambda_j^S(\cdot,a)] = -\mathbf{1}(j \leq k)(\hat{\lambda}_j^S(X,a) - \lambda_j^S(X,a))\prod_{i\neq j}(1 - \lambda_j^S(X,a)) \tag{134}$$

$$= -(\hat{\lambda}_j^S(X,a) - \lambda_j^S(X,a))\frac{\mathbf{1}(j \leq k)S_k(X,a)}{1 - \lambda_j^S(X,a)} \tag{135}$$

and also that

$$D_{\lambda_j^S(\cdot,a)}\xi_S(Z_a,\eta_t)[\hat{\lambda}_j^S(\cdot,a) - \lambda_j^S(\cdot,a)] \tag{136}$$

$$=(\hat{\lambda}_j^S(X,a) - \lambda_j^S(X,a))\sum_{i=0}^{t} -\frac{\mathbf{1}(i=j,\widetilde{T}\geq i)}{S_i(X,a)G_{i-1}(X,a)} \tag{137}$$

$$+ \frac{\left(\mathbf{1}(\widetilde{T}=i,\Delta^S=1) - \mathbf{1}(\widetilde{T}\geq i)\lambda_i^S(X,A)\right)\mathbf{1}(j\leq i)}{S_i(X,a)G_{i-1}(X,a)(1 - \lambda_j^S(X,a))} \tag{138}$$

$$=(\hat{\lambda}_j^S(X,a) - \lambda_j^S(X,a))\left(-\frac{\mathbf{1}(\widetilde{T}\geq j)}{S_j(X,a)G_{j-1}(X,a)} + \sum_{i=j}^{t}\frac{\left(\mathbf{1}(\widetilde{T}=i,\Delta^S=1) - \mathbf{1}(\widetilde{T}\geq i)\lambda_i^S(X,A)\right)}{S_i(X,a)G_{i-1}(X,a)(1 - \lambda_j^S(X,a))}\right) \tag{139}$$

$$=(\hat{\lambda}_j^S(X,a) - \lambda_j^S(X,a))\left(-\frac{\mathbf{1}(\widetilde{T}\geq j)}{S_j(X,a)G_{j-1}(X,a)} + \frac{\widetilde{\xi}_S(Z_a,\eta_t)}{1 - \lambda_j^S(X,a)}\right). \tag{140}$$

Here, it holds that $\mathbb{E}[\widetilde{\xi}_S(Z_a,\eta_t) \mid X, A] = 0$ following the same arguments as Lemma H.2.

For the cross-term, we obtain that

$$D_{\lambda_j^S(\cdot,a)}\xi_G(Z_a,\eta_{t-1})[\hat{\lambda}_j^S(\cdot,a) - \lambda_j^S(\cdot,a)] \tag{141}$$

$$=(\hat{\lambda}_j^S(X,a) - \lambda_j^S(X,a))\sum_{i=0}^{t-1}\frac{\left(\mathbf{1}(\widetilde{T}=i,\Delta^G=1) - \mathbf{1}(\widetilde{T}\geq i)\lambda_i^G(X,A)\right)\mathbf{1}(j\leq i-1)}{S_{i-1}(X,a)G_i(X,a)(1 - \lambda_j^S(X,a))} \tag{142}$$

$$=(\hat{\lambda}_j^S(X,a) - \lambda_j^S(X,a))\left(\frac{\bar{\xi}_G(Z_a,\eta_{t-1})}{1 - \lambda_j^S(X,a)}\right) \tag{143}$$

with $\mathbb{E}[\bar{\xi}_G(Z_a,\eta_{t-1}) \mid X, A] = 0$.

Using these calculation, we show orthogonality w.r.t. survival hazard functions via

$$D_{\lambda_j^S(\cdot,1)}D_g\mathcal{L}_f(g,\eta_t)[\hat{g} - g, \hat{\lambda}_j^S(\cdot,1) - \lambda_j^S(\cdot,1)] \tag{144}$$

$$=\frac{\mathrm{d}}{\mathrm{d}t}\left[D_g\mathcal{L}_f(g,\cdots,\lambda_j^S(\cdot,1) + t(\hat{\lambda}_j^S(\cdot,1) - \lambda_j^S(\cdot,1)))[\hat{g} - g]\right]_{t=0} \tag{145}$$

$$=\frac{-2}{\mathbb{E}[f(\widetilde{\eta}_t(X))]}\mathbb{E}\left[(\hat{g}(X) - g(X))(\hat{\lambda}_j^S(X,1) - \lambda_j^S(X,1))\left\{\frac{\partial f(\widetilde{\eta}_t(X))}{\partial \lambda_j^S(\cdot,1)}\left(S_t(X,1) - S_t(X,0) - g(X)\right.\right.\right. \tag{146}$$

$$\left. - \frac{A}{\pi(X)}S_t(X,1)\xi_S(Z,\eta_t) + \frac{1-A}{1-\pi(X)}S_t(X,0)\xi_S(Z,\eta_t)\right) + f(\widetilde{\eta}_t(X))\left[\frac{-S_t(X,1)}{1 - \lambda_j^S(X,1)}\right.\right. \tag{147}$$

$$\left. - \frac{A}{\pi(X)}\left(\frac{-S_t(X,1)\xi_S(Z,\eta_t)}{1 - \lambda_j^S(X,1)} + S_t(X,1)\left(\frac{-\mathbf{1}(\widetilde{T}\geq j)}{S_j(X,1)G_{j-1}(X,1)} + \frac{\widetilde{\xi}_S(Z_a,\eta_t)}{1 - \lambda_j^S(X,1)}\right)\right)\right] \tag{148}$$

$$+ \left[\frac{\partial^2 f(\widetilde{\eta}_t(X))}{\partial \lambda_j^S(\cdot,1)\partial\pi}(A - \pi(X)) - \frac{A}{\pi(X)}\left(\frac{\partial^2 f(\widetilde{\eta}_t(X))}{\partial \lambda_j^S(\cdot,1)\partial S_{t-1}(\cdot,1)}S_{t-1}(X,1)\xi_S(Z,\eta_t)\right.\right. \tag{149}$$

$$\left.\left. + \frac{\partial f(\widetilde{\eta}_t(X))}{\partial S_{t-1}(\cdot,1)}\left(\frac{-S_{t-1}(X,1)\xi_S(Z,\eta_{t-1})}{1 - \lambda_j^S(X,1)} + S_{t-1}(X,1)\left(\frac{-\mathbf{1}(\widetilde{T}\geq j)}{S_j(X,1)G_{j-1}(X,1)} + \frac{\widetilde{\xi}_S(Z_a,\eta_{t-1})}{1 - \lambda_j^S(X,1)}\right)\right)\right.\right. \tag{150}$$

$$+ \frac{\partial^2 f(\widetilde{\eta}_t(X))}{\partial \lambda_j^S(\cdot, 1) \partial G_{t-1}(\cdot, 1)} G_{t-1}(X, 1) \xi_G(Z, \eta_{t-1}) + \frac{\partial f(\widetilde{\eta}_t(X))}{\partial G_{t-1}(\cdot, 1)} G_{t-1}(X, 1) \left( \frac{\bar{\xi}_G(Z_a, \eta_{t-1})}{1 - \lambda_j^S(X, a)} \right) \Bigg) \Bigg] \tag{151}$$

$$(S_t(X, 1) - S_t(X, 0) - g(X))\} ] \tag{152}$$

$$+ \frac{2 D_g \mathcal{L}_f(g, \eta_t)[\hat{g} - g]}{\mathbb{E}[f(\widetilde{\eta}_t(X))]^2} \mathbb{E} \left[ \frac{\partial f(\widetilde{\eta}_t(X))}{\partial \lambda_j^S(\cdot, 1)} \right] \tag{153}$$

$$\overset{(*)}{=} \frac{-2}{\mathbb{E}[f(\widetilde{\eta}_t(X))]} \mathbb{E} \left[ (\hat{g}(X) - g(X))(\hat{\lambda}_j^S(X, 1) - \lambda_j^S(X, 1)) \left\{ \frac{\partial f(\widetilde{\eta}_t(X))}{\partial \lambda_j^S(\cdot, 1)} (S_t(X, 1) - S_t(X, 0) - g(X)) \right. \right. \tag{154}$$

$$+ f(\widetilde{\eta}_t(X)) \left[ \frac{-S_t(X, 1)}{1 - \lambda_j^S(X, 1)} - \frac{A}{\pi(X)} \left( S_t(X, 1) \left( \frac{-\mathbf{1}(\widetilde{T} \geq j)}{S_j(X, 1) G_{j-1}(X, 1)} \right) \right) \right] \tag{155}$$

$$+ \left[ -\frac{A}{\pi(X)} \frac{\partial f(\widetilde{\eta}_t(X))}{\partial S_{t-1}(\cdot, 1)} S_{t-1}(X, 1) \left( \frac{-\mathbf{1}(\widetilde{T} \geq j)}{S_j(X, 1) G_{j-1}(X, 1)} \right) \right] (S_t(X, 1) - S_t(X, 0) - g(X)) \Bigg\} \Bigg] \tag{156}$$

$$= \frac{-2}{\mathbb{E}[f(\widetilde{\eta}_t(X))]} \mathbb{E} \left[ (\hat{g}(X) - g(X))(\hat{\lambda}_j^S(X, 1) - \lambda_j^S(X, 1)) \right. \tag{157}$$

$$\left\{ \frac{\partial f(\widetilde{\eta}_t(X))}{\partial S_{t-1}(\cdot, 1)} \frac{\partial S_{t-1}(X, 1)}{\partial \lambda_j^S(\cdot, 1)} (S_t(X, 1) - S_t(X, 0) - g(X)) \right. \tag{158}$$

$$+ f(\widetilde{\eta}_t(X)) \left[ \frac{-S_t(X, 1)}{1 - \lambda_j^S(X, 1)} + \left( S_t(X, 1) \left( \frac{1}{1 - \lambda_j^S(X, 1)} \right) \right) \right] \tag{159}$$

$$+ \left[ -\frac{\partial f(\widetilde{\eta}_t(X))}{\partial S_{t-1}(\cdot, 1)} \frac{\partial S_{t-1}(X, 1)}{\partial \lambda_j^S(\cdot, 1)} \right] (S_t(X, 1) - S_t(X, 0) - g(X)) \Bigg\} \Bigg] \tag{160}$$

$$= 0, \tag{161}$$

where in $(*)$ we applied the law of total expectation and Lemma H.2 to remove all terms including any of $\xi_S(Z, \eta_t)$, $\widetilde{\xi}_S(Z, \eta_t)$, $\bar{\xi}_G(Z, \eta_{t-1})$, or $A - \pi(X)$ and the same argument to show that $D_g \mathcal{L}_f(g, \eta_t)[\hat{g} - g] = 0$. We can apply an analogous argument for $\lambda_j^S(\cdot, 0)$ and obtain

$$D_{\lambda_j^S(\cdot, 0)} D_g \mathcal{L}_f(g, \eta_t)[\hat{g} - g, \hat{\lambda}_j^S(\cdot, 0) - \lambda_j^S(\cdot, 0)] \tag{162}$$

$$= \frac{\mathrm{d}}{\mathrm{d}t} \left[ D_g \mathcal{L}_f(g, \cdots, \lambda_j^S(\cdot, 0) + t(\hat{\lambda}_j^S(\cdot, 0) - \lambda_j^S(\cdot, 0)))[\hat{g} - g] \right]_{t=0} \tag{163}$$

$$= 0, \tag{164}$$

which proves orthogonality w.r.t. survival hazard functions.

To show orthogonality w.r.t. censoring hazard functions. note that

$$D_{\lambda_j^G(\cdot, a)} G_k(X, a)[\hat{\lambda}_j^G(\cdot, a) - \lambda_j^G(\cdot, a)] = -(\hat{\lambda}_j^G(X, a) - \lambda_j^G(X, a)) \frac{\mathbf{1}(j \leq k) G_k(X, a)}{1 - \lambda_j^G(X, a)} \tag{165}$$

and also that

$$D_{\lambda_j^G(\cdot, a)} \xi_G(Z_a, \eta_{t-1})[\hat{\lambda}_j^G(\cdot, a) - \lambda_j^G(\cdot, a)] \tag{166}$$

$$= (\hat{\lambda}_j^G(X, a) - \lambda_j^G(X, a)) \left( -\frac{\mathbf{1}(\widetilde{T} \geq j)}{G_j(X, a) S_{j-1}(X, a)} + \frac{\widetilde{\xi}_G(Z_a, \eta_{t-1})}{1 - \lambda_j^G(X, a)} \right) \tag{167}$$

with $\mathbb{E}[\widetilde{\xi}_G(Z_a, \eta_{t-1}) \mid X, A] = 0$ following the same arguments as Lemma H.2. Furthermore,

$$D_{\lambda_j^G(\cdot, a)} \xi_S(Z_a, \eta_t)[\hat{\lambda}_j^G(\cdot, a) - \lambda_j^G(\cdot, a)] = (\hat{\lambda}_j^G(X, a) - \lambda_j^G(X, a)) \left( \frac{\bar{\xi}_S(Z_a, \eta_t)}{1 - \lambda_j^G(X, a)} \right) \tag{168}$$

with $\mathbb{E}[\bar{\xi}_S(Z_a, \eta_t) \mid X, A] = 0$.

Hence, for the second-order directional derivative w.r.t. censoring survival functions, we obtain

$$D_{\lambda_j^G(\cdot,1)} D_g \mathcal{L}_f(g, \eta_t)[\hat{g} - g, \hat{\lambda}_j^G(\cdot, 1) - \lambda_j^G(\cdot, 1)] \tag{169}$$

$$= \frac{\mathrm{d}}{\mathrm{d}t} \left[ D_g \mathcal{L}_f(g, \cdots, \lambda_j^G(\cdot, 1) + t(\hat{\lambda}_j^G(\cdot, 1) - \lambda_j^G(\cdot, 1)))[\hat{g} - g] \right]_{t=0} \tag{170}$$

$$= \frac{-2}{\mathbb{E}[f(\widetilde{\eta}_t(X))]} \mathbb{E} \left[ (\hat{g}(X) - g(X))(\hat{\lambda}_j^G(X, 1) - \lambda_j^G(X, 1)) \left\{ \frac{\partial f(\widetilde{\eta}_t(X))}{\partial \lambda_j^G(\cdot, 1)} \left( S_t(X, 1) - S_t(X, 0) - g(X) \right. \right. \right. \tag{171}$$

$$\left. - \frac{A}{\pi(X)} S_t(X, 1) \xi_S(Z, \eta_t) + \frac{1 - A}{1 - \pi(X)} S_t(X, 0) \xi_S(Z, \eta_t) \right) \tag{172}$$

$$+ f(\widetilde{\eta}_t(X)) \left[ -\frac{A}{\pi(X)} \left( S_t(X, 1) \left( \frac{\bar{\xi}_S(Z_a, \eta_t)}{1 - \lambda_j^G(X, 1)} \right) \right) \right] \tag{173}$$

$$+ \left[ \frac{\partial^2 f(\widetilde{\eta}_t(X))}{\partial \lambda_j^G(\cdot, 1) \partial \pi} (A - \pi(X)) - \frac{A}{\pi(X)} \left( \frac{\partial^2 f(\widetilde{\eta}_t(X))}{\partial \lambda_j^G(\cdot, 1) \partial S_{t-1}(\cdot, 1)} S_{t-1}(X, 1) \xi_S(Z, \eta_{t-1}) \right. \right. \tag{174}$$

$$+ \frac{\partial f(\widetilde{\eta}_t(X))}{\partial S_{t-1}(\cdot, 1)} \left( S_{t-1}(X, 1) \frac{\bar{\xi}_S(Z_a, \eta_{t-1})}{1 - \lambda_j^G(X, 1)} \right) + \frac{\partial^2 f(\widetilde{\eta}_t(X))}{\partial \lambda_j^G(\cdot, 1) \partial G_{t-1}(\cdot, 1)} G_{t-1}(X, 1) \xi_G(Z, \eta_{t-1}) \tag{175}$$

$$+ \frac{\partial f(\widetilde{\eta}_t(X))}{\partial G_{t-1}(\cdot, 1)} \left( \frac{-G_{t-1}(X, 1) \xi_G(Z, \eta_{t-1})}{1 - \lambda_j^G(X, 1)} + G_{t-1}(X, 1) \left( \frac{-\mathbf{1}(\widetilde{T} \geq j)}{G_j(X, 1) S_{j-1}(X, 1)} + \frac{\widetilde{\xi}_G(Z_a, \eta_{t-1})}{1 - \lambda_j^G(X, 1)} \right) \right) \right) \right] \tag{176}$$

$$\left. \cdot (S_t(X, 1) - S_t(X, 0) - g(X)) \right\} \right] + \frac{2 D_g \mathcal{L}_f(g, \eta_t)[\hat{g} - g]}{\mathbb{E}[f(\widetilde{\eta}_t(X))]^2} \mathbb{E} \left[ \frac{\partial f(\widetilde{\eta}_t(X))}{\partial \lambda_j^G(\cdot, 1)} \right] \tag{177}$$

$$\overset{(*)}{=} \frac{-2}{\mathbb{E}[f(\widetilde{\eta}_t(X))]} \mathbb{E} \left[ (\hat{g}(X) - g(X))(\hat{\lambda}_j^G(X, 1) - \lambda_j^G(X, 1)) \left\{ \frac{\partial f(\widetilde{\eta}_t(X))}{\partial \lambda_j^G(\cdot, 1)} (S_t(X, 1) - S_t(X, 0) - g(X)) \right. \right. \tag{178}$$

$$+ \left[ -\frac{A}{\pi(X)} \frac{\partial f(\widetilde{\eta}_t(X))}{\partial G_{t-1}(\cdot, 1)} G_{t-1}(X, 1) \left( \frac{-\mathbf{1}(\widetilde{T} \geq j)}{G_j(X, 1) S_{j-1}(X, 1)} \right) \right] (S_t(X, 1) - S_t(X, 0) - g(X)) \right\} \right] \tag{179}$$

$$= \frac{-2}{\mathbb{E}[f(\widetilde{\eta}_t(X))]} \mathbb{E} \left[ (\hat{g}(X) - g(X))(\hat{\lambda}_j^G(X, 1) - \lambda_j^G(X, 1)) \left\{ \frac{\partial f(\widetilde{\eta}_t(X))}{\partial G_{t-1}(\cdot, 1)} \frac{\partial G_{t-1}(X, 1)}{\partial \lambda_j^G(\cdot, 1)} (S_t(X, 1) - S_t(X, 0) - g(X)) \right. \right. \tag{180}$$

$$+ \left[ -\frac{\partial f(\widetilde{\eta}_t(X))}{\partial G_{t-1}(\cdot, 1)} \frac{\partial G_{t-1}(X, 1)}{\partial \lambda_j^G(\cdot, 1)} \right] (S_t(X, 1) - S_t(X, 0) - g(X)) \right\} \right] \tag{181}$$

$$= 0, \tag{182}$$

where in $(*)$ we applied the law of total expectation and Lemma H.2 to remove all terms including any of $\xi_S(Z, \eta_t)$, $\widetilde{\xi}_S(Z, \eta_t)$, $\bar{\xi}_G(Z, \eta_{t-1})$, or $A - \pi(X)$ and the same argument to show that $D_g \mathcal{L}_f(g, \eta_t)[\hat{g} - g] = 0$. Finally, we can apply the same line of arguments to $\lambda_j^G(\cdot, 0)$ to show

$$D_{\lambda_j^G(\cdot,0)} D_g \mathcal{L}_f(g, \eta_t)[\hat{g} - g, \hat{\lambda}_j^G(\cdot, 0) - \lambda_j^G(\cdot, 0)] \tag{183}$$

$$= \frac{\mathrm{d}}{\mathrm{d}t} \left[ D_g \mathcal{L}_f(g, \cdots, \lambda_j^G(\cdot, 0) + t(\hat{\lambda}_j^G(\cdot, 0) - \lambda_j^G(\cdot, 0)))[\hat{g} - g] \right]_{t=0} \tag{184}$$

$$= 0, \tag{185}$$

which completes the proof. $\qquad \square$

## H.3 Proof of Theorem 5.2

*Proof.* We write the orthogonal loss as

$$\mathcal{L}_f(g, \eta_t) = \frac{1}{\mathbb{E}[f(\widetilde{\eta}_t(X))]} \mathbb{E} \left[ \rho(Z, \eta_t) \left( \varphi(Z, \eta_t) - g(X) \right)^2 \right] \tag{186}$$

$$= \frac{1}{\mathbb{E}[f(\widetilde{\eta}_t(X))]} \mathbb{E}\left[\rho(Z,\eta_t)\left(\varphi(Z,\eta_t) - \tau_t(X) + \tau_t(X) - g(X)\right)^2\right] \tag{187}$$

$$= \frac{1}{\mathbb{E}[f(\widetilde{\eta}_t(X))]}\left(\mathbb{E}\left[\rho(Z,\eta_t)\left(\varphi(Z,\eta_t) - \tau_t(X)\right)^2\right] + 2\mathbb{E}\left[\rho(Z,\eta_t)\left(\varphi(Z,\eta_t) - \tau_t(X)\right)\tau_t(X)\right]\right) \tag{188}$$

$$+ \frac{1}{\mathbb{E}[f(\widetilde{\eta}_t(X))]}\left(-2\mathbb{E}\left[\rho(Z,\eta_t)\left(\varphi(Z,\eta_t) - \tau_t(X)\right)g(X)\right] + \mathbb{E}\left[\rho(Z,\eta_t)\left(\tau_t(X) - g(X)\right)^2\right]\right), \tag{189}$$

where only the terms in the last equation depend on $g$. The first term we can rewrite as

$$\mathbb{E}\left[\rho(Z,\eta_t)\left(\varphi(Z,\eta_t) - \tau_t(X)\right)g(X)\right] \tag{190}$$

$$= \mathbb{E}\left[\rho(Z,\eta_t)\left(S_t(X,1) - S_t(X,0) - \frac{(A - \pi(X))\xi_S(Z,\eta_t)S_t(X,A)f(\widetilde{\eta}_t(X))}{\pi(X)(1-\pi(X))\rho(Z,\eta_t)} - \tau_t(X)\right)g(X)\right] \tag{191}$$

$$= -\mathbb{E}\left[\mathbb{E}\left[\frac{(A - \pi(X))\xi_S(Z,\eta_t)S_t(X,A)f(\widetilde{\eta}_t(X))}{\pi(X)(1-\pi(X))}g(X)\Big| X, A\right]\right] \tag{192}$$

$$= -\mathbb{E}\left[\frac{(A - \pi(X))\mathbb{E}[\xi_S(Z,\eta_t) \mid X, A]S_t(X,A)f(\widetilde{\eta}_t(X))}{\pi(X)(1-\pi(X))}g(X)\right] \tag{193}$$

$$\overset{(*)}{=} 0, \tag{194}$$

where $(*)$ follows from Lemma H.2. For the second term, note that

$$\mathbb{E}\left[\rho(Z,\eta_t) \mid X\right] \tag{195}$$

$$= \mathbb{E}\left[f(\widetilde{\eta}_t(X)) + \frac{\partial f}{\partial \pi}(\widetilde{\eta}_t(X))(A - \pi(X))\right. \tag{196}$$

$$\left. - \left(\frac{A}{\pi(X)} + \frac{1-A}{1-\pi(X)}\right)\left(\frac{\partial f(\widetilde{\eta}_t(X))}{\partial S_{t-1}(\cdot, A)}S_{t-1}(X,A)\xi_S(Z,\eta_{t-1})\right.\right. \tag{197}$$

$$\left.\left. + \frac{\partial f(\widetilde{\eta}_t(X))}{\partial G_{t-1}(\cdot, A)}G_{t-1}(X,A)\xi_G(Z,\eta_{t-1})\right)\Big| X\right] \tag{198}$$

$$= f(\widetilde{\eta}_t(X)) + \frac{\partial f}{\partial \pi}(\widetilde{\eta}_t(X))(\pi(X) - \pi(X)) \tag{199}$$

$$- \frac{\pi(X)}{\pi(X)}\left(\frac{\partial f(\widetilde{\eta}_t(X))}{\partial S_{t-1}(\cdot, 1)}S_{t-1}(X,1)\mathbb{E}[\xi_S(Z,\eta_{t-1}) \mid X, A = 1]\right. \tag{200}$$

$$\left. + \frac{\partial f(\widetilde{\eta}_t(X))}{\partial G_{t-1}(\cdot, 1)}G_{t-1}(X,1)\mathbb{E}[\xi_G(Z,\eta_{t-1}) \mid X, A = 1]\right) \tag{201}$$

$$- \frac{1-\pi(X)}{1-\pi(X)}\left(\frac{\partial f(\widetilde{\eta}_t(X))}{\partial S_{t-1}(\cdot, 0)}S_{t-1}(X,0)\mathbb{E}[\xi_S(Z,\eta_{t-1}) \mid X, A = 0]\right. \tag{202}$$

$$\left. + \frac{\partial f(\widetilde{\eta}_t(X))}{\partial G_{t-1}(\cdot, 0)}G_{t-1}(X,0)\mathbb{E}[\xi_G(Z,\eta_{t-1}) \mid X, A = 0]\right) \tag{203}$$

$$\overset{(*)}{=} f(\widetilde{\eta}_t(X)), \tag{204}$$

where $(*)$ follows from Lemma H.2. Hence,

$$\mathbb{E}\left[\rho(Z,\eta_t)\left(\tau_t(X) - g(X)\right)^2\right] = \mathbb{E}\left[\mathbb{E}\left[\rho(Z,\eta_t)\left(\tau_t(X) - g(X)\right)^2 \mid X\right]\right] \tag{205}$$

$$= \mathbb{E}\left[\mathbb{E}\left[f(\widetilde{\eta}_t(X))\left(\tau_t(X) - g(X)\right)^2 \mid X\right]\right] \tag{206}$$

$$= \mathbb{E}\left[f(\widetilde{\eta}_t(X))\left(\tau_t(X) - g(X)\right)^2\right]. \tag{207}$$

Putting everything together, we obtain

$$\mathcal{L}_f(g,\eta_t) = \frac{1}{\mathbb{E}[f(\widetilde{\eta}_t(X))]}\left(\mathbb{E}\left[\rho(Z,\eta_t)\left(\varphi(Z,\eta_t) - \tau_t(X)\right)^2\right]\right. \tag{208}$$

$$\left. + 2\mathbb{E}\left[\rho(Z,\eta_t)\left(\varphi(Z,\eta_t) - \tau_t(X)\right)\tau_t(X)\right]\right) \tag{209}$$

$$+ \frac{1}{\mathbb{E}[f(\widetilde{\eta}_t(X))]} \mathbb{E}\left[ f(\widetilde{\eta}_t(X)) \left( \tau_t(X) - g(X) \right)^2 \right], \tag{210}$$

which proves the claim because the first two summands do not depend on $g$ and do not affect the minimization. $\qquad\square$

# I Implementation details

## I.1 Data generation

**Synthetic data generation:** We generated two different scenarios for our experiments on synthetic data, from which we each generated four different datasets with full overlap, low treatment overlap, low censoring overlap, and low survival overlap, respectively. For **Scenario 1**, we sample a one-dimensional confounder from a standard normal distribution and set $T = 5$. Then we generate the propensities as well as the censoring and survival hazard for the full overlap setting as

$$\pi(x) = 0.5\sigma(x) + 0.2\sigma(-x) \tag{211}$$

$$\lambda_t^G(x, a) = 0.5\sigma(x + t) \tag{212}$$

$$\lambda_t^S(x, a) = 0.5\sigma(x - t), \tag{213}$$

where $\sigma$ denotes the sigmoid function. For the low overlap settings, we then replace $\pi(x), \lambda_t^G(x, a)$ and $\lambda_t^S(x, a)$ by

$$\pi(x) = \sigma(2x) \tag{214}$$

$$\lambda_t^G(x, a) = \sigma(1.5(x + t)) \tag{215}$$

$$\lambda_t^S(x, a) = \sigma(-ax/(t + 1)), \tag{216}$$

respectively.

For the more difficult **Scenario 2**, we follow [8] and sample a ten-dimensional standard normal confounder. In this scenario, the full overlap setting with $T = 30$ is generated as

$$\pi(\mathbf{x}) = \sigma\left(\sum(\mathbf{x})\right) \tag{217}$$

$$\lambda_t^G(\mathbf{x}, a) = 0 \tag{218}$$

$$\lambda_t^S(\mathbf{x}, a) = \begin{cases} 0.1\sigma\left(-0.5\left(\sum\mathbf{x}\right)^2\right), & t \le 10, \\ 0.1\sigma\left(10\left(\sum\mathbf{x}\right)^2\right), & t > 10. \end{cases} \tag{219}$$

To generate the different low-overlap settings, we then update the functions by

$$\pi(\mathbf{x}) = \sigma(3x_0) \tag{220}$$

$$\lambda_t^G(\mathbf{x}, a) = 0.1\sigma\left(10\left(\sum\mathbf{x}\right) + at\right) \tag{221}$$

$$\lambda_t^S(\mathbf{x}, a) = \begin{cases} 0.1\sigma\left(-0.5\left(\sum\mathbf{x}\right)^2 - a(0.5 + \mathbb{1}\{\sum\mathbf{x} \ge 0\})\right), & t \le 10, \\ 0.1\sigma\left(10\left(\sum\mathbf{x}\right)^2 - a(0.5 + \mathbb{1}\{\sum\mathbf{x} \ge 0\})\right), & t > 10, \end{cases} \tag{222}$$

respectively. From all datasets, we generate 30000 train samples with a train-validation split of 0.6 and 3000 test samples. For Scenario 1, we evaluate our model on time steps 0 to 5, and for Scenario 2 on time steps 0,5,10, and 15.

**Twins data:** For our medical case study, we employ the Twins dataset as in [31]. The dataset considers the birth weight of 11984 pairs of twins born in the USA between 1989 and 1991 with respect to mortality in the first year of life. Treatment $a = 1$ corresponds to being born the heavier twin. The dataset further contains 46 confounders associated with the parents, the pregnancy, and the birth. For a detailed description of the dataset, see [31]. We follow the pre-processing steps and censoring mechanism in [8] to create the time-to-event outcome.

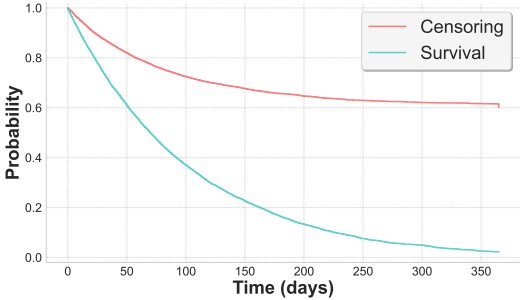

Figure 7: Survival and censoring curves over time averaged across all samples.

In Figure 7, we assess the survival and censoring overlap averaged across all samples. Overall, there is no sign of systematic low survival or censoring overlap on average, i.e., the mean-aggregated curves across all covariate realizations are bounded away from zero.

### I.2 Implementation:

All experiments were run in Python and run on an AMD Ryzen 7 PRO 6850U 2.70 GHz CPU with eight cores and 32GB RAM. Our experiments can be easily computed on standard computing resources. We provide our code at `https://github.com/m-schroder/OrthoSurvLearners`

**Model architecture and parameters:** Throughout our experiments, we instantiate all models with the *same* four-layer neural network architectures and hyperparameters. This allows us to assess the effect of our proposed weighting scheme, as differences in performance can be merely attributed to the different orthogonal loss functions for training the second-stage model.

For the synthetic datasets, our propensitiy networks consist of 20 hidden neurons per layer and are trained over 10 epochs with batch size 64, learning rate 0.0001 dropout factor 0.1. Our hazard networks were trained without droput across 40 epochs and with batch size 256. Finally, the second-stage models, consisting of 64 neurons per layer, were trained without droput across 30 epochs with batch size 64 and learning rate 0.00001.

For the real-world dataset, we adapted the training of the propensity network to 20 epochs and the training of the hazard networks to 120 epochs. We increased the hidden dimension of the second stage model to 32 neurons per layer and wich was trained over 300 epochs with learning rate 0.000001. All other parameter specifications remained as for the synthetic datasets.

# J  Additional experiment

## J.1  Ablation study

Here, we provide additional empirical results by evaluating our method across varying sample sizes using synthetic Dataset 1 with low censoring overlap (see Tab. 4). We report the results for the C and the T+C learner. PEHE results are averaged over the time steps to the power of $10^2$. As expected, the variance increases with smaller sample sizes, but overall, the proposed learners remain robust to changes in dataset size.

Table 4: Results by learner and dataset size (PEHE $\pm$ standard deviation).

| Learner | 50% dataset | 100% dataset | 150% dataset |
|---|---|---|---|
| C | $0.2541 \pm 1.0848e^{-3}$ | $0.0836 \pm 8.1418e^{-5}$ | $0.0571 \pm 3.0890e^{-5}$ |
| S | $0.0722 \pm 3.1591e^{-5}$ | $0.0677 \pm 1.0696e^{-5}$ | $0.0535 \pm 1.0176e^{-5}$ |
| T+C | $0.2227 \pm 1.2506e^{-3}$ | $0.1100 \pm 2.2593e^{-5}$ | $0.0637 \pm 4.6489e^{-5}$ |
| T+S | $0.0720 \pm 3.0045e^{-5}$ | $0.1602 \pm 1.0559e^{-5}$ | $0.0533 \pm 1.0353e^{-5}$ |

## J.2  ADJUVANT dataset

The ADJUVANT trial [30, 55] enrolled 171 patients with EGFR-mutant stage II–IIIa non-small cell lung cancer (NSCLC). Baseline characteristics such as age, sex, and smoking history were recorded, and comprehensive sequencing of 422 cancer-related genes (including CDK4 and MYC) was performed. The trial's primary objective was to compare the efficacy of adjuvant gefitinib versus chemotherapy.

While previous analyses of the ADJUVANT trial relied on linear survival models, we demonstrate how nonlinear survival learners provide new insights. Specifically, we observed more pronounced effects on disease-free survival (DFS) time in patient subgroups with co-alterations in TP53 and SMAD4 (average subgroup effect of 0.2087 vs. total ATE of 0.1882), co-alterartions TP53 and NKX2-1 (average subgroup effect of 0.1997) or who are of 65 years and older (average subgroup effect of 0.2052) across all learners. Additionally, patient subgroups with co-amplification of TP53 and SMAD4 (average subgroup effect of 0.2075 vs. total ATE of 0.1870) or of old age (average subgroup effect of 0.2040) showed a benefit in overall survival (OS) time when treated with gefitinib .

