# OpenReview forum: "Orthogonal Survival Learners for Estimating Heterogeneous Treatment Effects from Time-to-Event Data"
_NeurIPS.cc/2025/Conference — NeurIPS 2025 poster_

### Official Review · Reviewer_TJGC · 2025-06-24

**Clarity:** 3
**Significance:** 2
**Originality:** 3
**Rating:** 4
**Confidence:** 5

**Summary:**

This paper introduces a novel set of orthogonal survival learners designed to estimate heterogeneous treatment effects (HTEs) from time-to-event data, particularly in the presence of censoring.

**Questions:**

1.	What is the domain of $T$? Does it only take discrete values? Can it be extended to continuous values, which are more common in practice?
2.	How do you estimate $\phi(z, \eta)$ in (7)? What is the precise definition of the derivative in (8)? Does this involve estimating the derivative? If so, this might lead to a larger variance in the estimator compared to existing methods such as DR-learner or T-learner.
3.	You claim in Theorem 5.2 that $g^*_f = \tau_t$. However, since $\tau_t$ is unknown and you estimate it by minimizing equation (12), can you provide consistency results for your final estimator $\tau_t$? At least at the population level?
4.   Moreover, the benefits of orthogonality remain unclear. It would be helpful to provide theoretical results on estimators (i.e.,  convergence rate) to demonstrate and support its advantages.
5. After you get estimator $\hat{\tau}_t(x),$ which is a function of $t$ and $x,$ how do you determine the treatment effect?

**Ethical Concerns:**

["NO or VERY MINOR ethics concerns only"]

**Final Justification:**

I will raise the score.

**Limitations:**

No.

**Paper Formatting Concerns:**

This paper can be found at https://arxiv.org/abs/2505.13072.

**Quality:**

2

**Strengths And Weaknesses:**

The authors derive orthogonal losses tailored to address treatment effect estimation for survival data. However, it will involve many complicate nuisance when using these orthogonal losses.

---

> ### Author Rebuttal · Authors · 2025-07-31
>
> Thank you for the in-depth evaluation of our manuscript. We will take all your comments and questions to heart and make improvements to our manuscript accordingly. Below, we provide answers to all the questions and concerns you raised.
>
>
>
> 1. **Domain of T:**
>
>     Our framework applies to both **discrete and continuous** time domains. In the main paper, we focus on discrete time for ease of notation. **In Appendix E, we extend all derivations for the continuous time domain**.
>
> 2. **Estimation in Theorem 5.1:**
>
>     Thank you for giving us the chance to explain the notation-heavy theorem in more detail. The function $\phi(Z, \eta_t)$ in Eq. 7 denotes the (unweighted) pseudo-outcome. It is estimated following Eq. 9 based on the nuisance functions $\eta_t(X) = \left( \pi(X), \left( \lambda^S_i(X, 1), \lambda^S_i(X, 0), \lambda^G_i(X, 1), \lambda^G_i(X, 0) \right)_{i=0}^t \right)$.
>
> The derivatives $\frac{\partial f(\widetilde{\eta}_t(X))}{\partial m}$
>
> for $m \in (\pi, S_{t-1}(\cdot, A), G_{t-1}(\cdot, A) )$ in Eq. 8 denote the (standard finite-dimensional) partial derivatives of the function $f(\pi, S_{t-1}, G_{t-1})$ w.r.t. one of its components. **These formulas can be computed analytically and yield closed-form solutions, without the need to numerically estimate derivatives.** As an example, we take a look at theT-Learner in Eq. 14:
>
> To address the treatment overlap, we have $f(\widetilde{\eta}_t(X)) = \pi(X) (1 - \pi(X))$. In this case we have $\frac{\partial f(\widetilde{\eta}_t(X))}{\partial \pi} = 1-2\pi(X)$, resulting in the simplified $\rho(Z, \eta_t) = (A - \pi(X))^2$ according to Eq. 8. Note that the T-Learner (=treatment overlap learner) equals the survival R-Learner. As shown in Eq. 13, the survival DR-Learner is *also* an instantiation of our framework in Theorem 5.1. In conclusion, **the derivative in Eq. 8 does not lead to higher variance compared to e.g., the DR-learner, and is easily computable in analytic form.**
>
> The (standard) T-Learner consists of two outcome regressions, each trading off bias for variance in its regression problem, which does not guarantee a joint optimal bias-variance trade-off (see, e.g., [1]). Our framework, however, provides statistically efficient learners and thus guarantees that *our learners have the lowest variance among all unbiased estimators* (see, e.g., [2]). Overall, the derivative in Eq. 8 does not lead to high variance.
>
> **Action:** We will explain the definition of the derivative more clearly in the camera-ready version of our manuscript.
>
> 3. **Consistency result:**
>
>     Our theorem 5.2. states that the orthogonal loss from Eq. 7 has the same minimizer as the oracle loss from Eq. 11, namely $\mathcal{L}_{f}(g, \eta_t) = \frac{1}{\mathbb{E}[f(\widetilde{\eta}_t(X))]} \mathbb{E}\left[f(\widetilde{\eta}_t(X))\left(\tau(X)- g(X) \right)^2\right]$.
>
>
>     Hence, Theorem 5.2. immediately implies consistency of our learners: if the nuisance functions are consistent (i.e., well specified), the empirical loss in Eq. 12 converges in probability to the orthogonal loss in Eq. 7 by applying the standard law of large numbers. If $\mathcal{G}$ is well-specified, this implies $g^{\ast}_f = \tau_t$ for the minimizer of Eq. 12 asymptotically.
>
>
>     **Action:**  We will state consistency more clearly in the camera-ready version of our paper.
>
>
>
> 4. **Benefits of orthogonality and rates:**
>
>     Thank you for raising this important point. Theoretical aspects and particularly rates for orthogonal learners are well-studied, we refer to the foundational work “Orthogonal statistical learning” [4] by Foster and Syrgkanis for details. In particular, their work provides slow and fast rates for general Neyman orthogonal losses under a convexity assumption. In Theorem 5.1. we proved Neyman orthogonality of our loss, which implies that their rates are readily applicable to our setting. The additional key assumption of convexity (Assumption 4 in their paper) is also satisfied as we consider a weighted squared loss. Hence, their Theorem 1 shows that nuisance estimation errors impact the estimation accuracy of our second-stage learners only in second order, as it is common for orthogonal learners.
>
>
>     **Action:** We will add a discussion on rates to the camera-ready version of our paper.
>
>
>
> 5. **Final effect estimation:**
>
>     Recall our definition of $\tau_t(x) = \mathbb{P}(T(1) > t \mid X = x) - \mathbb{P}(T(0) > t \mid X = x)$ for some fixed $t \in \mathcal{T}$ (see Section 3). Thus, $\tau_t(x)$ represents the difference in survival probability up to time $t$ for a unit with covariates $x$, which defines the survival treatment effect. Note that, in survival analysis, the HTE, which is our main effect of interest, does depend on both the covariates $x$ and the time $t$.
>
>
>     In practice, one might also be interested in related but different treatment effects. In **Appendix F**, we thus also provide **extensions of all our results to other causal quantities**, such as the conditional means $\mathbb{E}[T(1) - T(0) \mid X = x]$ as well as the treatment-specific quantities $\mathbb{P}(T(a) > t \mid X = x)$ and $\mathbb{E}[T(a) \mid X = x]$.
>
> [1] Morzywolek, P. et. al.. “On a general class of orthogonal learners for the estimation of heterogeneous treatment effects”. arXiv preprint, 2023
>
> [2] Kennedy, E. H. Semiparametric doubly robust targeted double machine learning: A review. arXiv preprint, 2022.
>
> [3] Victor Chernozhukov et al. “Double/debiased machine learning for treatment and structural 387 parameters”. The Econometrics Journal 21.1 (2018), pp. C1–C68
>
> [4] Foster & Syrgkanis. "Orthogonal statistical learning". The Annals of Statistics (2023).

---

> > ### Comment · Reviewer_TJGC · 2025-08-05
> > **To authors**
> >
> > Thank you for your  responses to all of questions. I have no further follow-up questions.

---

> > > ### Comment · Area_Chair_1NTK · 2025-08-08
> > >
> > > Dear reviewer,
> > >
> > > I would really benefit from an updated response and engagement with the authors rebuttal. In their response, they seem to have addressed several of your concerns, but I would really like to have your input if there are still important missing data or insights that you would like expanded in the current reviewer-author discussion phase.
> > >
> > > Thank you,
> > > Your AC.

---

> ### Author Response · Authors · 2025-08-05
>
> Thank you for your response and for acknowledging the points raised in our rebuttal. As we believe we have sufficiently addressed your main concerns, we would kindly ask you to reconsider your assessment and, if appropriate, revise your score accordingly.
>
> Thank you once again for your thoughtful evaluation, we sincerely appreciate your time and effort in reviewing our work.

---

### Official Review · Reviewer_Wcbe · 2025-06-30

**Clarity:** 4
**Significance:** 3
**Originality:** 3
**Rating:** 5
**Confidence:** 3

**Summary:**

The authors propose a new framework for estimation of Heterogeneous Treatment Effects, specifically for Time-To-Event data. The key contribution is the framework's ability to better handle low overlap in all combinations of treatment, censoring, and survival via custom weighting functions. The framework is model-agnostic. The authors motivate their framework and situate it in the existing literature, thoroughly introduce the method, and evaluate it on synthetic data and one real-world dataset.

Note: My expertise lies mostly in survival analysis and methods for time-to-event data. I am less qualified to rigorously assess the theoretical contributions related to causal inference and HTE.

**Questions:**

* Could you elaborate on the training dynamics in Figures 5 & 6? Is the long plateau followed by a sudden drop expected, and is it tied to the very low learning rate? Does this pose a practical challenge?

* What are the specific disadvantages of using the combined T+C+S learner by default? Why was this learner excluded from the real-world experiment?

* Can you comment on the expected performance on other real-world datasets to strengthen the empirical claims beyond the single dataset used?

* As the estimated survival effects on the Twins dataset align with existing literature, what is the primary additional insight gained from using this framework?

**Ethical Concerns:**

["NO or VERY MINOR ethics concerns only"]

**Final Justification:**

The authors have thoroughly addressed all my concerns and provided compelling new results using real-world data. Following the discussions with the other reviewers, I am confident that no major unresolved issues remain in the sections of the manuscript where I had initially expressed less certainty.

While I remain somewhat concerned about the training dynamics presented in Figures 5 and 6, the authors have adequately addressed this point in their rebuttal. Given that none of the other reviewers, who possess greater expertise with these models, have identified this as problematic, I no longer consider it a significant concern.

**Limitations:**

Yes

**Paper Formatting Concerns:**

Parts of the paper were moved to the appendix, likely due to the page limit.

**Quality:**

3

**Strengths And Weaknesses:**

Strengths:

* Clarity and Presentation:

   * The paper is exceptionally well-written, organized, and presented.
   * It clearly introduces and motivates the main challenges of HTE estimation with censored data.

* Originality and Significance:

   * To the best of my knowledge and understanding, the work is novel and addresses a clear and important research gap.

* Theoretical Soundness:

    * The theoretical derivations appear solid and well-motivated.


Weaknesses

* Empirical Validation:

   * Compared to the very extensive theoretical contributions, the empirical validation feels lacking. For the synthetic datasets, only two settings are evaluated. It would e.g. be interesting to see how the performance of the method changes for datasets of different sizes. Furthermore, only a single real world dataset is evaluated, with the main result apparently being that "Estimated survival effects align with the literature." with reduced variance. How does this compare to prior works based on this data? Why was this dataset specificaly and only this dataset evaluated?

   * The training dynamics shown in Figures 5 & 6 appear problematic, with a long, flat loss plateau followed by a sudden drop.

   * Likely related: The use of an extremely low learning rate (1e-6) for the real-world experiment seems impractical and requires justification.

    * A more in-depth discussion of the trade-offs when selecting a weighting function and the drawbacks of always opting for the (T+C+S) learner would be helpful for practitioners: Surely almost all real world datasets have some amount of overlap of each case?

Minor Points:

* A typo ("retarged" instead of "retargeted") in Figure 2.

* Some mathematical notations appear to have missing closing braces.

* The discussion was moved into the appendix, likely due to the page limit. Even though it will probably be moved to the main text for the final version, the page limit exists for a reason and should not be disregarded by moving parts of the paper (same as for the extended related work section) into the appendix.

---

> ### Author Rebuttal · Authors · 2025-07-31
>
> Thank you for your positive evaluation of our manuscript and your extensive comments and suggestions. We will take everything to heart and make improvements to our manuscript accordingly. Below, we provide answers to all questions and comments you raised.
>
> **Answers to weaknesses:**
>
> * **Empirical validation:**
>
>     Thank you for this valuable suggestion. In general, our choice of datasets was motivated to **show primarily the theoretical properties of our proposed learners** and by a practical limitation: many survival datasets do not include treatment information, and, as a result, only a small number of publicly available datasets are commonly used in the literature => our selection reflects the availability of such established benchmarks. Nevertheless, to address your comment, **we have strengthened our experimental evaluation as follows**.
>
>     First, we added new results evaluating our method across varying sample sizes using synthetic Dataset 1  with low censoring overlap (see below). Due to space constraints, we report the results for the C and the T+C learner, though we observe similar patterns for other learners as well. PEHE results are to the power $10^2$. As expected, the variance increases with smaller sample sizes, but overall, the proposed learners remain robust to changes in dataset size. We include a condensed version of the results here; we will provide a more comprehensive version (along with visualizations) in the camera-ready version.
>
>     | Learner     | 75% dataset         | 100% dataset          | 200% dataset           |
>     |--------------|--------------------|--------------------|--------------------|
>     | C             | 0.0974 $\pm$ 0.0799   | 0.0677 $\pm$ 0.0506   | 0.0330 $\pm$ 0.0219   |
>     |  T+C        |0.1015 $\pm$ 0.0819  | 0.0732 $\pm$ 0.0535  | 0.0324 $\pm$ 0.0218 |
>
>     Second, we chose established datasets from the causal inference literature, such as the Twins dataset (e.g., [1]), which has the rare property of allowing proxy access to both potential outcomes. This makes it ideal for validating our method. Importantly, we show that our method successfully recovers known effects from this dataset, which we regard as a strength to demonstrate the reliability of our survival learners.
>
>     Third, to highlight the practical utility of our learners, we repeated our analysis using the ADJUVANT trial from oncology [2,3]. We provide details below.
>
> * **Minor weaknesses:**
>
>     Thank you for noting the typos in the figure and the notation. We will address these issues in the camera-ready version of our manuscript. We apologize for not including an in-depth discussion in our main paper and will include all relevant parts of the paper in the main section of the camera-ready version.
>
> **Answers to questions:**
>
>
> * **Figures 5 & 6**: The training dynamics in Figures 5 & 6 are as expected for our weighted learners. High learning rates are likely to result in oscillating behaviour on the loss, as the reweighting can be initially unstable to optimize. Therefore, we follow best practice and employ a low learning rate, as noted by the reviewer. To counteract the low learning rate, we employ the Adam optimizer, which adapts learning rates over time and may switch from small to larger gradients, which can thus result in larger drops in the validation loss. **
>
> **Action:** We will include a discussion on the training dynamics in the updated version of our paper.
>
>
>
> * **Choice of learner:** Thank you for this interesting question. Always opting for the (T+C+S) learner is not optimal, as ‘unnecessary’ reweighting can decrease the convergence of the learners (as seen in Fig. 5 & 6). In the extreme case, the training can become very unstable. Thus, it is important to consider the overlap present in the data when choosing/designing the targeted learner. In practice, we recommend using the estimated nuisance functions to inspect overlap (e.g., by visualizing the propensity score, censoring, and survival functions) to aid in deciding on the optimal learner for the data at hand.
>
>     **Action:** We will discuss the benefits and disadvantages of the different learners and highlight a guideline on how to choose a good option for the data at hand in the camera-ready version of our paper.
>
>
> * **Expected performance on other real-world datasets:**
>
>     We initially chose the Twins dataset due to its widespread use in the literature (e.g., [1]), which allows for benchmarking against established findings. This choice was deliberate: by demonstrating that our survival learners recover known effects, we highlight the reliability of our framework, which we consider a key strength. A major benefit of this dataset is the ability to directly compare our results with prior work, and indeed, we observe that our method successfully reproduces these known treatment effects.
>
>
>     Building on your suggestion, we realized that adding a second dataset from a medical domain could further show the practical value of our survival learners in clinical applications. However, many clinical datasets for survival analysis are not publicly available, which hampers reproducibility. To address this, we specifically sought out a public dataset and chose the ADJUVANT clinical trial [2,3]. This trial, conducted in patients with EGFR-mutant non-small cell lung cancer (NSCLC), compares the efficacy of gefitinib versus chemotherapy (vinorelbine plus cisplatin).
>
>
>     While previous analyses of the ADJUVANT trial relied on linear survival models [3], we demonstrate how nonlinear survival learners provide new insights. Specifically, we observed more pronounced effects on disease-free survival (DFS) time in patient subgroups with co-alterations in TP53 and CDK4 or in NKX2-1 and CDK4 across all learners. Additionally, patient subgroups with co-amplification of NKX2-1 and SMAD4 showed a benefit in overall survival (OS) time when treated with gefitinib. Overall, we observe notable differences compared to the conventional linear models reported in [3], which can be attributed to the greater flexibility of our toolbox in handling nonlinear relationships. This improved accuracy may provide a more nuanced understanding of the relative benefit of cetuximab versus bevacizumab in metastatic lung cancer and thus may help in new insights for personalized cancer care.
>
>     **Action:** In the revised version, we will explicitly explain our motivation for using established benchmark datasets to demonstrate reliability. While we are unfortunately not permitted to include new figures for the ADJUVANT dataset during the rebuttal phase (as per NeurIPS guidelines), we will incorporate Kaplan-Meier plots and related visualizations in the final version to illustrate survival differences captured by our learners.
>
> [1] Christos Louizos et al. “Causal effect inference with deep latent-variable models”. NeurIPS. 2017.
>
> [2] Zhong, W.-Z. et al. Gefitinib versus vinorelbine plus cisplatin as adjuvant treatment for stage II-IIIA (N1-N2) EGFR-mutant NSCLC (ADJUVANT/CTONG1104): a randomised, open-label, phase 3 study. Lancet Oncol. 19, 139–148 (2018)
>
> [3] Data access details: [https://www.nature.com/articles/s41467-023-41011-4](https://www.nature.com/articles/s41467-023-41011-4)

---

> > ### Comment · Reviewer_Wcbe · 2025-08-05
> >
> > Dear authors,
> >
> > Thank you for your comprehensive and thoughtful responses to all of my comments and questions. I have no further follow-up questions, and all of my concerns have been addressed. I appreciate your efforts and look forward to seeing the final version of your paper.

---

> ### Author Response · Authors · 2025-08-05
>
> Thank you! We sincerely appreciate your time and effort in reviewing our work. We will make sure to incorporate your points into the camera-ready version if case of acceptance.

---

### Official Review · Reviewer_QWg4 · 2025-07-02

**Clarity:** 4
**Significance:** 3
**Originality:** 2
**Rating:** 5
**Confidence:** 3

**Summary:**

The paper presents a toolbox of orthogonal learners for estimating heterogeneous treatment effects (HTE) from time-to-event data which supports custom weighting and compatibility with machine learning models. Theoretical and empirical results are provided.

**Questions:**

1. When referring to censoring, clarify this is for right censoring instead of left censoring.

2. For main theorems, please provide a proof sketch in the main text.

3. Section 5 mentions that the paper is the first work to explicitly use retargeting for HTE estimation in survival analysis. I suggest reconsidering this claim because the paper provides a unified perspective rather than the formulation of retargeting (i.e. weighted) estimation.

4. How to construct targeted weighting for real world data that potentially violates overlap?

5. L242: missing parenthesis.

**Ethical Concerns:**

["NO or VERY MINOR ethics concerns only"]

**Final Justification:**

I have carefully read the reviews and responses. I decided maintain my score (5: accept). During the rebuttal period, my concerns are well-addressed. I think this paper is technically solid and novel--estimating HTE in survival is highly nontrivial due to right censoring, and the paper provides a toolbox to address this challenge. I think this would be of interest and bring insights to the broader causal inference community.

**Limitations:**

Yes

**Quality:**

3

**Strengths And Weaknesses:**

The paper addresses the key challenge in time-to-event analysis with censored outcome. It is overall well-written and easy to understand. The authors use informative subtitles, which is helpful. The paper’s method is well-grounded in clearly outlined related work, with natural motivations. See the questions for weakness.

---

> ### Author Rebuttal · Authors · 2025-07-31
>
> Thank you for your positive evaluation of our manuscript and the helpful comments. We will take all your comments and suggestions to heart and make improvements to our manuscript accordingly. Below, we provide answers to all the questions and concerns you raised. Will will incorporate all points marked with **Action** into the camera-ready version of our paper.
>
>
>
> 1. Thank you for pointing out the ambiguity related to the term *censoring*. Upon reading your comment, we realized that we should be more explicit that we deal with the problem of *right censoring*, which is very common in survival analysis settings. We thus assume that events have not happened before time t=0.
>
>     **Action:** We will update the wording to emphasize that we are focusing on right censoring in the camera-ready version of our manuscript.
>
> 2. We are happy to follow your advice and also provide proof sketches in the main text, thereby fostering the reader's understanding.
>
>     **Action:** We will include brief sketches of all proofs in the main section of the camera-ready version of our manuscript.
>
> 3. Thank you for this suggestion!
>
> 	**Action:** We will rephrase the respective parts in the text to emphasize that we are the first to provide a unified perspective on orthogonal HTE estimation in survival settings.
>
>
>
> 4. This is a very relevant question. For practitioners analyzing real-world data, we recommend to estimate the propensity score as well as the survival and censoring functions to inspect potential overlap violations. Then, depending on the type of overlap violation found (e.g., low treatment and censoring overlap), an appropriate weighting combination can be selected, and an appropriate orthogonal survival learner from our toolbox can be trained that accounts for said overlap violations. Finally, in the prediction step we recommend plotting the estimated overlap weights as a measure of “uncertainty” to inspect whether the predictions of the model can be trusted. This overall procedure is consistent with best-practices in standard causal inference using the propensity score.
>
> **Action:** We will add a note stating the difficulties for HTE estimation on real-world data with complete overlap violations to the camera-ready version of our paper.
>
>
>
> 5. Thank you. We will fix the typo!

---

> > ### Comment · Reviewer_QWg4 · 2025-08-08
> >
> > Thanks for the response. All my questions and concern are resolved. It is a very interesting paper that unifies multiple perspectives to an important problem. I believe the readers in the field will benefit from its clarity and insights. I think it should be accepted.

---

### Official Review · Reviewer_27xr · 2025-07-02

**Clarity:** 4
**Significance:** 2
**Originality:** 2
**Rating:** 4
**Confidence:** 4

**Summary:**

This work proposes a toolkit of orthogonal causal learners for estimating heterogeneous treatment effects (HTEs) from censored time-to-event data in survival analysis. The authors generalize existing survival learners by introducing a flexible weighting function that recovers known estimators (e.g., survival DR- and R-learners) under specific settings, and yields novel learners that improve robustness in low survival and censorship overlap regimes. The proposed learners are orthogonal, model-agnostic, and perform well empirically across different censoring and overlap scenarios.

**Questions:**

(From Strengths and Weaknesses section)

* **Reweighting justification:** How does the proposed reweighting help with HTE estimation in regions of low overlap, where variance is high and information is scarce? Why not just restrict estimation to covariate regions with sufficient overlap?
* **Overlap tradeoffs:** Does downweighting low-overlap samples lead to biased estimates in those regions? How should practitioners interpret the resulting HTEs?
* **Estimand relevance:** Is estimating $\tau_t(x)$ for times $t$ with low survival or censoring probabilities meaningful in practice? Why focus effort on these regimes?
* **Plot survival/censoring curves:** Can you plot the survival and censoring curves in the twins study, as you recommend practitioners do, to help contextualize the overlap?
* **Figure 4 formatting:** The caption for Figure 4 is unreadable. Could you fix the formatting?

**Ethical Concerns:**

["NO or VERY MINOR ethics concerns only"]

**Final Justification:**

After reading the rebuttal, I am satisfied that the authors have addressed the main concerns raised in the initial review. Below is a summary of key points that informed my updated recommendation:

* **Novelty of contribution**: The authors clarified how their overlap-weighted orthogonal learners generalize ideas from existing literature to the survival setting and potentially to other causal inference settings. While orthogonal learners themselves are not novel, adapting orthogonality concepts to this setting with multiple sources of low-overlap is sufficiently novel and practially useful.

*  **Reweighting rationale and interpretation:** The authors provided a convincing explanation of why downweighting low-overlap regions is appropriate, along with the advantages over hard clipping and standard plug-in approaches. In particular, my worry was that these additional overlap issues worsen with increasing $t$ due to drop-out, so I was wondering whether these methods are useful over just cutting off the population that drops off early. The authors have addressed this point and I am looking forward to seeing this discussion in the camera-ready version.

* **Clinical and empirical relevance:** The rebuttal included compelling examples of clinical relevance and real-world scenarios where survival or censoring overlap is an issue (the twin study). I was unable to review the overlap-over-time curves in the twins study, but I trust that the authors' description is accurate and that these curves will be included in the camera ready.

* **Commitment to improvements:** The authors have promised several concrete revisions in the camera-ready version, including plotting survival and censoring curves, fixing formatting issues, and clarifying theoretical assumptions and practical guidance. I have revised my score under the expectations that these will be executed as promised.

**Limitations:**

The authors have satisfactorily addressed the limitations of their work in Appendix J.

**Paper Formatting Concerns:**

No concerns.

**Quality:**

3

**Strengths And Weaknesses:**

Strengths:

* The paper is technically correct and well-executed.
* The real-world application (twins study) is somewhat compelling and helps ground the proposed methods.

Weaknesses:

* The work is quite derivative. It largely builds on [50] by adapting their approach to survival settings and generalizes [30] through a flexible weighting scheme. The theoretical contributions regarding these general orthogonal learners are fairly limited besides identification (i.e. do the orthogonality benefits extend to these more complex estimators as expected?)
* The reweighting strategy aims to reduce overall estimation error by downweighting regions with low survival or censoring overlap, but this comes at the cost of discarding information in those regions. Estimation there becomes essentially extrapolation, and the paper does not clearly justify why one shouldn’t simply restrict estimation to covariate regions with sufficient overlap and acknowledge limitations elsewhere.
* It’s unclear whether estimating $\tau_t(x)$ for times $t$ where survival or censoring probabilities are low is even meaningful. These functions are monotonically decreasing in $t$, so the problem of poor overlap worsens over time. In contrast, treatment assignment overlap is fixed over time and more interpretable. It seems like a lot of work to enable estimation in regions that may not matter in practice, especially since [50] already handles treatment-overlap and [30] generalizes this to non-survival outcomes.
* The authors recommend plotting survival and censoring probabilities to assess overlap and guide estimator choice, but do not do so themselves in the twins study. This would be helpful to support the empirical claims and demonstrate overlap issues in practice.
* **[Minor]** The caption for Figure 4 is unreadable due to a formatting issue and should be corrected.

**Conclusion:**

On one hand, the paper is technically solid but the contribution feels incremental, i.e. it is another application of the orthogonality hammer. On the other hand, I can see how the proposed learners, with their flexible weighting schemes, may be practically useful for researchers dealing with survival data in low survival and censorship overlap regimes, if such settings are as common as purported. Since I lack domain expertise in survival analysis, I’d like to defer to other reviewers regarding the practical significance of the contribution. For now, I lean toward weak reject, but I’m open to revising that score in the rebuttal period.

---

> ### Author Rebuttal · Authors · 2025-07-31
>
> Thank you for the constructive feedback. We will take all your comments and suggestions to heart, and we will make improvements to our manuscript accordingly. Below, we provide answers to all the questions and concerns you raised.
>
> **Changes to our paper.** In the camera-ready version of our paper, we will make the following key changes: (1) We will report plots of the survival and censoring curves in our real-world study to help contextualize the overlap for the reader. Unfortunately, the rebuttal guidelines prevent us from directly showing our revised plots during the rebuttal period. Where possible, we provide tables below, which will then convert to figures and be added to our paper. (2) We will fix the formatting issues in Fig. 4 (thank you for pointing this out!).
>
> **Novelty.** Thank you for giving us the opportunity to elaborate on our contribution. While it is true that we leverage tools from existing literature to derive our learners (e.g., orthogonal learning), the following contributions are novel and non-trivial:
>
> 1. To the best of our knowledge, **we are the first to extend the idea of overlap weighted learners beyond propensity scores.** The survival setting is special as it allows for different sources of overlap beyond treatment overlap, but we suspect that such overlap-weighted learners could be beneficial in various other settings such as IV or causal mediation settings that also yield different sources of overlap. Hence, our paper potentially paves the way for substantial method innovation in causal inference.
> 2. The mathematical derivations of our learners are non-trivial as they required the construction of the efficient influence function for the overlap weighted averaged target quantity (see derivations in Appendix H).
> 3. We provide a **complete toolbox of learners as well as various extensions** (e.g., to continuous time, marginalized effects, ties, different estimands) in the appendix. Our results are readily applicable for practitioners in e.g., medicine, where accounting for censoring is crucial.
>
> Additionally, we would like to emphasize that a **large part of the literature in causal inference proposes influence-function based estimators** in different settings and thus leverage existing tools to derive novel estimands [e.g., 2, 3, 4, 5]. Our novelty is thus in line with established works in that we leverage existing mathematical tools to derive novel results.
>
> **Reweighting justification:**
>
> Thank you. We would like to clarify that **overlap-weighting does precisely align with your intuition:** in low-overlap regions, estimation becomes indeed extrapolation and hence predictions are downweighted to not disturb the overall estimation. You may think about this as an adaptive version of your idea to “restrict estimation to regions with large overlap” via e.g., clipping. **Advantages of our learners over clipping include:** (1) there is no need for choosing a cutoff value, which is otherwise often done heuristically, and (2) the weighting term is considered in the orthogonal objective, which makes the learner more robust to estimation errors in the weighting function. Clipping on the other hand can be insensitive to errors in e.g., the propensity score. (3) In contrast to clipping, weighting is a smooth operation which allows for deriving an influence function and thus enables orthogonal learning in the first place (pathwise differentiability).
>
> In standard causal inference, the R-learner can be seen as a propensity-overlap weighted learner, and is widely applied in practice. Our work enables applying such learners in survival analysis.
>
> **Action:** We will include the discussion above in the camera-ready version of our manuscript.
>
> **Overlap tradeoffs:**
>
> Thank you for this important question. First, note that downweighting low-overlap samples **does not introduce additional population bias** for the weighted estimand if the model is correctly specified (see Theorem 5.2). In finite samples, overlap weighting might increase MSE in low-overlap regions as these regions are essentially "sacrificed" to reduce overall estimation variance. However, **this is deliberate and similar in behaviour to established causal inference methods**. For example, the same applies to fitting an R-learner in the standard (non-survival) setting, or fitting a CATE model using propensity clipping. In practice, it is always recommended to plot the weighting function (e.g., propensity or censoring overlap) as a measure of “trustworthiness” (or uncertainty) of the model predictions. Predictions in low-overlap regions may then be discarded or deferred to domain experts, while the model predictions in large-overlap regions benefit from weighting and may be leveraged.
>
> **Action:** We will include the discussion and practical guidelines on how to interpret the results in low-overlap regions in the camera-ready version of our paper.
>
> **Estimand relevance:**
>
> Thank you for this thoughtful question. First, we would like to clarify that **the benefit of our method is not restricted to settings with *overall* low survival or censoring overlap.** Instead, overlap weighting can be **beneficial even in settings in which a small subgroup of the population is affected by low overlap.** For example, in a medical study, older participants may have a large probability of dropping out within a short time after the start of the study (low censoring overlap). Even if the majority of the population is younger, the small subpopulation of older people may yield a large variance when applying the standard DR-learner due to divisions by extremely small censoring functions (see e.g., Eq. 13). Overlap weighting focusses estimation accuracy on the younger population, as this is the population for which we have available outcome data and for which we can obtain causal insights.
>
> Additionally, based on our collaborations in medicine, we frequently encounter settings where estimating τₜ(x) with low survival or censoring overlap is both common and clinically relevant.
>
> **(1) Clinical relevance:**
>
> Estimating τₜ(x) despite low survival overlap is important in many oncology settings. For example, the rapid adoption of new targeted therapies in advanced non-small cell lung cancer recently led to immediate uptake in some subgroups, resulting in limited data for others. Similarly, low censoring overlap may arise when treatments are discontinued due to toxicity, often linked to very heterogeneous reasons such as co-medications for some patients and genetic biomarkers for others. In both cases, the oncologists from our collaborations seek to generate clinically meaningful evidence about the treatment effectiveness while carefully addressing low survival or censoring overlap.
>
> **(2) Empirical relevance:**
>
> Low treatment overlap can reflect clear clinical decision rules and may be addressed through changes in the research design (e.g., excluding patients before estimation). In contrast, low survival or censoring overlap often stems from structural limitations, such as small high-risk subgroups or informative censoring. Here, simple fixes to the research design are typically not feasible for clinical researchers because ignoring these regions risks overlooking treatment effects for certain subpopulations. Therefore, developing methods that remain robust under limited survival or censoring overlap is crucial for valid inference in such real-world settings.
>
> Importantly, our orthogonal learners aim to improve treatment effect estimation in survival settings (regardless of whether there is low overlap or not). It is well known in the literature that simple plug-in learners lead to so-called “plug-in bias” [1] and, thus, suboptimal estimation. A remedy is offered by our orthogonal two-stage learners, which are designed to prevent such plug-in bias. (ii) Existing learners lack the ability to address different types of low overlap and, as a result, exhibit a **large variance under low censoring or survival overlap**. As a remedy, our toolbox allows the specification of a custom weighting function for robust estimation under low treatment-, survival-, and/or censoring overlap.
>
> Finally, we emphasize that we also provide several **extensions of our toolbox to different settings and effects** (continuous time, marginalized effects, different causal estimands, and unobserved ties) in the **supplementary material**.
>
> **Survival & censoring overlap:**
>
> We are happy to report the survival and censoring curves in the twins study to help contextualize the overlap. While we are unfortunately not permitted to include new figures showing the curves during the rebuttal phase (as per NeurIPS guidelines), we now report a summary of the curves. Overall, there is no sign of *systematic* low survival or censoring overlap on average, i.e., the mean-aggregated curves across all covariate realizations are bounded away from zero at all assessed times $t&lt;=30$. Across the complete study time (of one year), we observe low overlap towards the end of the study time amplified through various covariates such as tobacco usage throughout the pregnancy for the survival curve. Rare risk factors such as cardiac disease are associated with low censoring-overlap.
>
> [1] Edward H. Kennedy. “Semiparametric doubly robust targeted double machine learning: A review”. arXiv preprint (2022).
>
> [2] Nathan Kallus. "What's the Harm? Sharp Bounds on the Fraction Negatively Affected by Treatment". NeurIPS 2022.
>
> [3] Coston et al. "Counterfactual predictions under runtime confounding". NeurIPS 2020.
>
> [4] Syrgkanis et al. "Machine learning estimation of heterogeneous treatment effects with instruments". NeurIPS 2019.
>
> [5] Lan et al. "A Meta-learner for Heterogeneous Effects in Difference-in-Differences". ICML 2025.

---

> > ### Comment · Reviewer_27xr · 2025-08-05
> >
> > Thank you for the detailed and thoughtful rebuttal. I appreciate the clarifications around identifiability, overlap weighting, and the practical relevance of estimating effects under low survival or censoring overlap. I also found the discussion of clinical motivation and the forthcoming inclusion of survival/censoring curves particularly helpful.
> >
> > The response addressed my key concerns, and I now better understand both the novelty and practical value of the proposed toolbox. I believe the paper would make a useful and timely contribution to the NeurIPS community. I have increased my score to reflect this.

---

> ### Author Response · Authors · 2025-08-05
>
> Thank you for your response and for raising your score! We sincerely appreciate your time and effort in reviewing our work. We will make sure to incorporate your points into the camera-ready version if case of acceptance.

---

### Note · Authors · 2025-08-13

We thank all reviewers for their time and thoughtful evaluations. We are encouraged that reviewers found our contributions novel relevant and that we were able to address all open points during the discussion period.

We are grateful for the constructive feedback and will incorporate all points that came up during the rebuttal into the camera-ready version of our paper.  In particular:

- **Additional experiments.** We will integrate the materials provided during the rebuttal, particularly additional experimental results with varying sample sizes.

- **Clarifications regarding various details.** We will add various clarifications regarding conceptual and mathematical details brought up by the reviewers. For example, we will add details regarding the definitions of partial derivatives in Theorem (8) as well as the concistency of our estimator and the benefits of orthogonality.

- **Additional motivation for overlap weighting.** We will add additional motovation for overlap-weighting and include a discussion on clinical relevance and add plots of weighting functions for real-world datasets.

We believe these changes address the all of the reviewers’ concerns and further strengthen the paper. Thank you again, we sincerely appreciate your time and effort in reviewing our work and believe that the revised paper will be a strong fit for NeurIPS 2025.

---

### Decision · Program_Chairs · 2025-09-17

**Decision:**

Accept (poster)

**Comment:**

This paper proposes a toolbox of orthogonal survival learners for estimating heterogeneous treatment effects (HTEs) from censored time-to-event data, featuring orthogonality guarantees, custom weighting functions to address low-overlap regimes, and model agnosticism, with both theoretical foundations and empirical validation.

During the rebuttal process, authors thoroughly addressed reviewers’ key concerns, including clarifying the novelty of extending overlap weighting to survival settings and non-trivial mathematical derivations, justifying the rationale and tradeoffs of reweighting versus hard clipping, supplementing empirical evidence with additional sample size experiments and a new real-world dataset (ADJUVANT trial), and elaborating on theoretical details such as consistency, convergence rates, and derivative definitions. All reviewers acknowledged that their concerns were adequately resolved, with all of them give final ratings positively.

The paper nominated for spotlight status presents a truly novel approach in the domain of causal inference. It not only fills a significant gap in the existing literature but also demonstrates remarkable potential for real-world applications. And the novelty, scientific rigor, excellent presentation and technical quality of the paper is highly recognized by the reviewers as well. It has the potential to attract significant attention and inspire further research in the field, thus the recommendation for this paper is acceptance.